# TS-Attn: Temporal-wise Separable Attention for Multi-Event Video Generation

**Hongyu Zhang**[1]*, **Yufan Deng**[1]*, **Zilin Pan**[1], **Peng-Tao Jiang**[2], **Bo Li**[2],
**Qibin Hou**[3], **Zhen Dong**[4] **Zhiyang Dou**[5] **Daquan Zhou**[1]†

[1]Peking University, Shenzhen Graduate School   [2]vivo BlueImage Lab   [3]Nankai University
[4]University of California, Santa Barbara   [5]The University of Hong Kong
{zhanghy@stu, dengyufan10@stu, daquan.zhou@}.pku.edu.cn

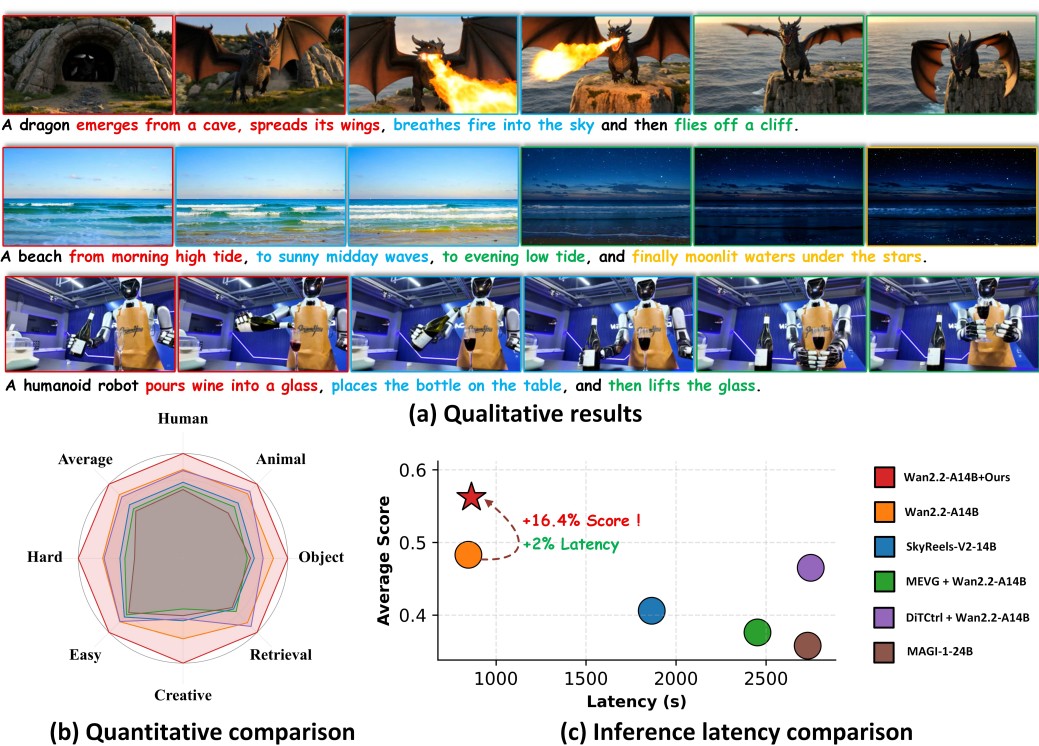

A dragon emerges from a cave, spreads its wings, breathes fire into the sky and then flies off a cliff.

A beach from morning high tide, to sunny midday waves, to evening low tide, and finally moonlit waters under the stars.

A humanoid robot pours wine into a glass, places the bottle on the table, and then lifts the glass.

**(a) Qualitative results**

**(b) Quantitative comparison**

**(c) Inference latency comparison**

Figure 1: We present **TS-Attn**, a training-free attention mechanism, which enhances multi-event video generation through alleviating attention conflicts across multi-event conditions. **(a)** Qualitative results across subjects and scenes. **(b)** Quantitative comparison on StoryEval-Bench. **(c)** Latency-performance tradeoff analysis.

## ABSTRACT

Generating high-quality videos from complex temporal descriptions that contain multiple sequential actions is a key unsolved problem. Existing methods are constrained by an inherent trade-off: using multiple short prompts fed sequentially into the model improves action fidelity but compromises temporal consistency, while a single complex prompt preserves consistency at the cost of prompt-following capability. We attribute this problem to two primary causes: 1) temporal misalignment between video content and the prompt, and 2) conflicting attention coupling between motion-related visual objects and their associated text conditions. To address these challenges, we propose a novel, training-free attention mechanism, **Temporal-wise Separable Attention** (TS-Attn), which dynamically

---

*Equal contribution.
†Corresponding author.

rearranges attention distribution to ensure temporal awareness and global coherence in multi-event scenarios. TS-Attn can be seamlessly integrated into various pre-trained text-to-video models, boosting StoryEval-Bench scores by 33.5% and 16.4% on Wan2.1-T2V-14B and Wan2.2-T2V-A14B with only a 2% increase in inference time. It also supports plug-and-play usage across models for multi-event image-to-video generation. The source code and project page are available at https://github.com/Hong-yu-Zhang/TS-Attn.

# 1 INTRODUCTION

Video generation models have undergone remarkable advancements, demonstrating impressive progress in their generation capabilities Blattmann et al. (2023a); Wang et al. (2024); Kong et al. (2024); Deng et al. (2026), which has in turn sparked a wide range of downstream applications Deng et al. (2025b); Hu et al. (2025); Deng et al. (2025a); Guo et al. (2024). Through the optimization of model architectures Peebles & Xie (2023); Flux (2024) and the scaling of training data Wang et al. (2025a), current models are capable of generating high-quality videos. However, the current good performance is mostly limited to the prompts containing single events, even for the state-of-the-art open-source models Wang et al. (2025a); Yang et al. (2024); Kong et al. (2024). How to faithfully generate videos from complex temporal descriptions (e.g., containing multiple events and dynamic motion information) remains underexplored.

Existing approaches can be broadly categorized into two streams, each facing inherent performance trade-offs. The first stream decomposes a complex multi-event prompt into several single-event prompts and executes them across multiple inference stages Lin et al. (2023); Zhang et al. (2024). While this paradigm is capable of producing action-rich content, combining individually generated clips using techniques such as KV cache Cai et al. (2025) or initial noise optimization Oh et al. (2024) often results in content drift and pronounced temporal inconsistencies Kim et al. (2025) with significantly increased inference time overhead. Conversely, the second stream of methods directly feeds the entire complex multi-event prompt into more powerful text encoders. Although this paradigm yields videos with improved consistency and global coherence Wang et al. (2025a); Zhang & Agrawala (2025), it often exhibits limited prompt-following ability, failing to accurately interpret and respond to all individual events. Such limitations frequently manifest as event omission or temporal hallucination.

Achieving an optimal trade-off requires simultaneously balancing global consistency and prompt adherence. ***This raises a key question: can we preserve global coherence with a single complex prompt while ensuring that the video accurately responds to each event in the correct temporal order?*** As illustrated in Figure 3, our analysis shows that the primary cause of weak prompt-following lies in temporal misalignment and the entangled attention correlations between motion-related regions of video tokens and the textual conditions of multiple events. To address this, we propose a simple yet effective idea: disentangle the video–text attention distribution associated with different events in the prompt and realign it with the corresponding individual events, ensuring that they remain separable along the temporal dimension with proper transitions.

From this observation, we derive two key insights: (1) motion-related regions in each frame should focus primarily on the event that occurs at the same time, and (2) interactions across different events in the temporal dimension should be minimized.

Building on the above insights, we propose **Temporal-wise Separable Attention (TS-Attn)**, a method that dynamically adjusts the attention distribution in the cross-attention layer to enable temporal awareness in multi-event scenarios. Our idea is intuitive: TS-Attn first extracts and thresholds the cross-attention map associated with the event-performing entity to identify the motion-related regions. TS-Attn then rearranges the attention distribution between motion-related video tokens and each event condition with proper separation to strengthen the correspondence with the temporally aligned event while reducing attention coupling from unrelated events. Finally, TS-Attn incorporates an attention reinforcement mechanism that adaptively scales event-related attention values based on the attention distribution: a smoother distribution indicates that more modifications are needed.

In summary, the key contributions of our work are as follows:

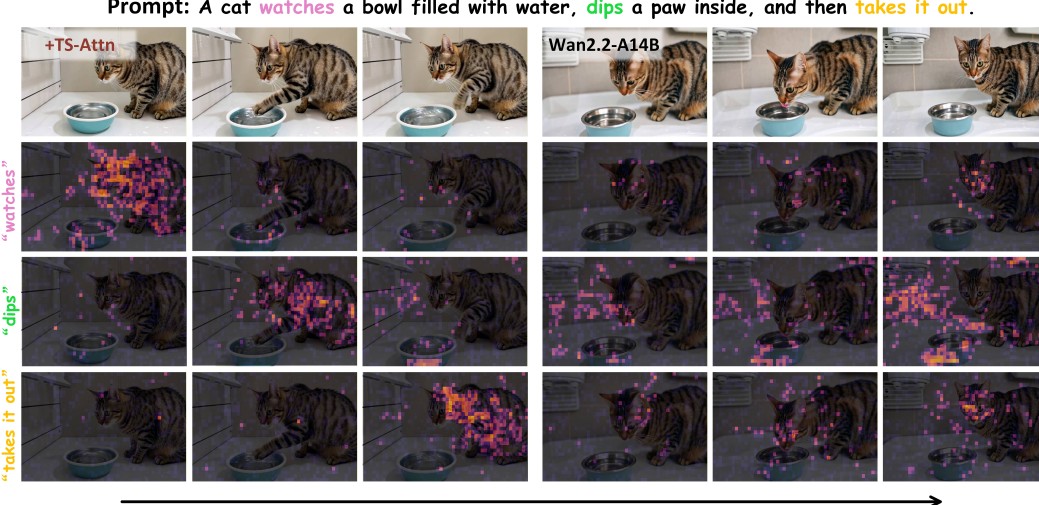

**Prompt:** A cat watches a bowl filled with water, dips a paw inside, and then takes it out.

Attention map along the temporal sequence

Figure 2: **Comparison of attention maps along the temporal sequence between TS-Attn and vanilla cross-attention.** TS-Attn strengthens motion-event alignment and reduces cross-event interference, ensuring accurate attention distribution among multiple events.

- We conduct an in-depth analysis of the root causes underlying poor prompt-following performance in complex descriptions, and reveal that temporally separable grouping is essential to prevent temporal conflicts.

- We propose a novel framework, TS-Attn, which dynamically restructures the attention distribution between motion-related regions and multi-event conditions. This design enables accurate event responses in the correct temporal order, while simultaneously preserving global consistency and ensuring physically plausible transitions.

- We conduct extensive experiments demonstrating that TS-Attn can be used in a training-free manner and seamlessly integrated into diverse video generation foundation models. Both qualitative and quantitative results show that it substantially improves baseline performance with negligible inference overhead, while remaining effective across multiple tasks, including multi-event text-to-video (T2V) and image-to-video (I2V).

## 2 RELATED WORKS

**Diffusion-based video generation.** Initial efforts concentrated on integrating temporal attention mechanisms into the 2D U-Net architecture Ronneberger et al. (2015), allowing image generation models to better capture the temporal dynamics required for video synthesis Blattmann et al. (2023b); Wang et al. (2023b); Khachatryan et al. (2023); Chen et al. (2024). As diffusion transformers (DiTs) gained prominence Ma et al. (2024), the focus shifted towards employing 3D full attention, effectively bridging spatial and temporal dependencies Zheng et al. (2024); Lin et al. (2024); Zhang et al. (2025a). This innovation laid the foundation for scalable models such as CogVideoX Yang et al. (2024), LTX-Video HaCohen et al. (2024), HunyuanVideo Kong et al. (2024), and Wan Wang et al. (2025a), which advanced the generation of high-resolution, temporally consistent video content.

**Multi-event video generation.** Several studies address multi-event video generation by breaking it into sequential multi-prompt generation Wang et al. (2023a); Qiu et al. (2024); Kim et al. (2025). MEVG Oh et al. (2024) ensures visual coherence by initializing each clip's noise with the inverted last frame of the previous clip, while DiTCtrl Cai et al. (2025) enables smooth motion transitions via mask-guided key–value sharing. However, these approaches require repeated inference, increasing computational costs and causing temporal inconsistencies. Another line of methods uses local and global cross-attention to strengthen responses to multiple sub-prompts Wang et al. (2025c); Tian

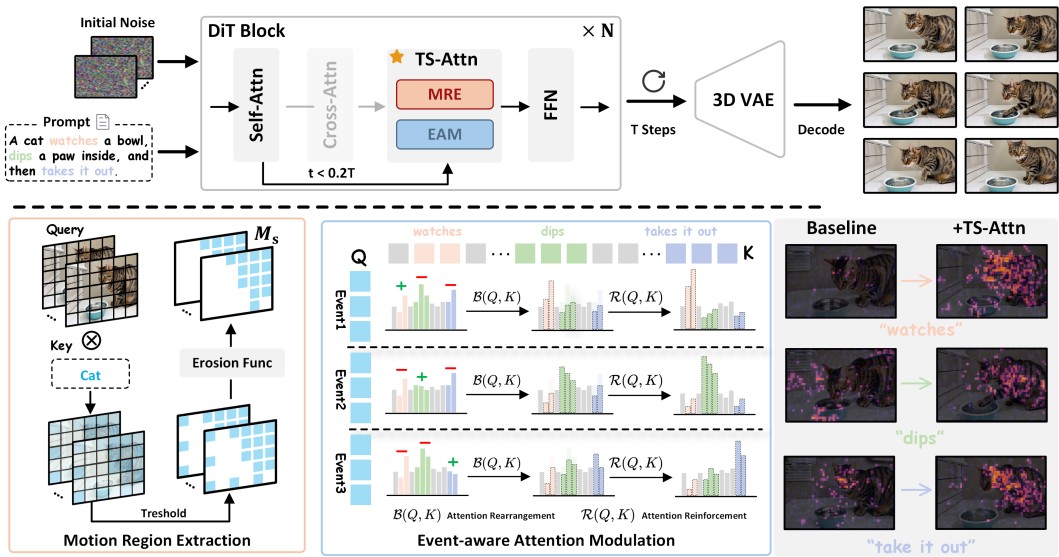

Figure 3: **The overall framework of TS-Attn.** TS-Attn replaces the original cross-attention in early denoising stages to incorporate motion information with temporal awareness. It consists of a motion region extraction module to identify motion-related tokens and an event-aware attention modulation module to adjust their attention distribution across multiple events.

et al. (2024); Bansal et al. (2024). However, the use of hard-masked attention Tian et al. (2024); Bansal et al. (2024) for overly strict control can lead to issues such as background inconsistency and makes it difficult to process fine-grained temporal transitions when foreground subjects are small.

To address this issue, recent approaches focus on packaging individual events into a global prompt for single-pass inference generation. Among them, MinT Wu et al. (2025b) and ShotAdapter Kara et al. (2025) rely on large amounts of timestamp-labeled data for post-training to enable the model to handle multi-event scenarios. However, this requires extensive computational resources and is difficult to adapt to new models. An intuitive approach is to use more powerful video generation foundation models, with features such as the ability to handle more complex prompts (e.g., Wan Wang et al. (2025a), HunyuanVideo Kong et al. (2024)) and longer frame durations (e.g., Framepack Zhang & Agrawala (2025), SkyReels-V2 Chen et al. (2025), MAGI-1 Teng et al. (2025)). Yet in practice, these models still struggle with complex multi-event prompts, often leading to event omissions or temporal coupling, underscoring the need for more robust solutions.

## 3 METHOD

### 3.1 INSIGHTS OF TS-ATTN

We conduct an in-depth analysis of why existing state-of-the-art foundation models encounter issues such as event omission and temporal errors when a single sentence contains multiple events. Specifically, we examine whether the keyframes of the generated video establish the correct temporal correspondence between video tokens and event conditions within the cross-attention layer. Since motion information is primarily formed in the early stages of denoising Zhang et al. (2025b), the middle layer at 20% of the denoising steps is used for attention analysis.

As illustrated on the right of Figure 2, we identify two critical issues in the cross-attention distribution of Wan2.2-A14B: (1) Motion-related regions (i.e., the layout of the subject "cat") in each frame fail to establish strong attention associations with their corresponding verbs in the temporal sequence. For instance, "watch" loosely aligns with the layout of "cat", while actions like "dips" and "take it out" focus on irrelevant background areas, leading to severe misalignment. (2) Attention coupling of verbs from different events occurs within the same frame. For example, in the middle frame, all three verbs exhibit strong responses on the same video regions.

The issues discussed above lead to incorrect injection of multiple event conditions, resulting in severe event omission and temporal errors. This phenomenon indicates that the cross-attention map requires significant recalibration to accommodate the temporal distribution of multiple events.

To address these issues, TS-Attn is designed based on two core insights: 1) Strengthen the attention correlation between each frame's motion-relevant region and its corresponding temporal event; 2) Minimize interference caused by coupled attention across different events. As expected, the implemented TS-Attn significantly improves temporal attention alignment across multiple events, ensuring faithful generation of multi-event sequences (Figure 2 left).

## 3.2 Overall Framework

As shown in Figure 3, the overall framework is implemented based on the DiT architecture. We replace the original cross-attention with TS-Attn in the early denoising stages to inject motion information with stronger temporal awareness. TS-Attn consists of two components: first, it identifies motion-region video tokens using the motion-related subject semantic layout, then applies event-aware attention modulation to these video tokens.

The temporal segmentation of multiple event intervals for video tokens can be simply achieved through various methods, including user input, leveraging efficient LLM APIs (e.g., GPT-4o-mini), or default uniform segmentation. These approaches show minimal differences in final performance. Details can refer to Appendix Table 8. By default, we use GPT-4o-mini for temporal segmentation, unless otherwise specified in the context.

## 3.3 Motion Region Extraction

To achieve precise attention modulation, TS-Attn first adaptively identifies motion regions across the video. Motion information in a video primarily originates from the foreground subject performing actions. Thus, the semantic layout of the subject in the prompt can approximately represent motion-related regions. As shown in Figure 3, given the query of the video tokens $Q \in \mathbb{R}^{N \times d}$ and the key of the text tokens $K \in \mathbb{R}^{M \times d}$, we obtain the semantic map $A_s \in \mathbb{R}^{N \times 1}$ of the subject $s$ :

$$A_s = \text{Mean}\left(\mathcal{I}_s\left(\frac{QK^\top}{\sqrt{d}}\right)\right), \tag{1}$$

where $\mathcal{I}_s(\cdot)$ represents the function for indexing subject $s$. Similar to Helbling et al. (2025), we use the mean value of $A_s$ as an adaptive threshold to obtain the motion region mask $M_s \in \mathbb{R}^{N \times 1}$:

$$M_s = \mathcal{F}_\mathcal{K}\left(\mathbb{I}\left(A_s \geq \text{Mean}(A_s)\right)\right), \tag{2}$$

where $\mathcal{F}_\mathcal{K}(\cdot)$ represents the erosion function with a kernel $\mathcal{K}$, which is used to remove scattered noise and refine the boundaries of the binary mask. Experimentally, $\mathcal{K}$ is set to 3. Finally, we can use $M_s$ to guide attention modulation in motion-related regions.

## 3.4 Event-aware Attention Modulation

To address the temporal misalignment and coupling of multi-events observed in Figure 2, event-aware attention modulation in TS-Attn is divided into two components: *attention rearrangement* and *attention reinforcement*.

*Attention rearrangement* is directly based on the insight from Sec. 3.1. It redistributes the attention in cross-attention along the temporal dimension, ensuring that motion-related video tokens in each frame focus on their temporally corresponding events while attenuating attention to other events. *Attention reinforcement* adaptively adjusts the intensity of attention based on the sharpness of the attention distribution, ensuring balanced and event-aware attention scaling. Therefore, the entire attention modulation process in TS-Attn can be formulated as follows:

$$A = \text{softmax}\left(\frac{QK^\top + M_s \odot \mathcal{R}(Q, K) \odot \mathcal{B}(Q, K)}{\sqrt{d}}\right) \in \mathbb{R}^{N \times M}, \tag{3}$$

where $\mathcal{B}(Q, K)$ is the bias function to achieve attention rearrangement, $\mathcal{R}(Q, K)$ is the attention reinforcement function, and $M_s$ is derived from Sec. 3.3 to constrain attention modulation in the motion-related region. The details of these two functions are introduced below separately.

**Attention Rearrangement.** Given the event token list $[\boldsymbol{e}_1, \boldsymbol{e}_2, \ldots, \boldsymbol{e}_m]$ in the prompt, and the corresponding temporally segmented video queries $[\boldsymbol{Q}_1, \boldsymbol{Q}_2, \ldots, \boldsymbol{Q}_m]$ as described in Sec. 3.2. Attention rearrangement encourages each temporally segmented video query to interact with its corresponding event while weakening the influence of other events. This process is mainly achieved by applying positive bias $\boldsymbol{b}_i^+$ or negative bias $\boldsymbol{b}_i^-$ to different events:

$$\boldsymbol{b}_i^+ = \max(\boldsymbol{Q}_i \boldsymbol{K}^\top) - \text{mean}(\boldsymbol{Q}_i \boldsymbol{K}^\top), \tag{4}$$

$$\boldsymbol{b}_i^- = \min(\boldsymbol{Q}_i \boldsymbol{K}^\top) - \text{mean}(\boldsymbol{Q}_i \boldsymbol{K}^\top), \tag{5}$$

$$\mathcal{B}(\boldsymbol{Q}_i, \boldsymbol{K})[x, y] = \begin{cases} \boldsymbol{b}_i^+[x, y], & \text{if } y \in e_i, \\ \boldsymbol{b}_i^-[x, y], & \text{if } y \in e_j, i \neq j \\ 0, & \text{otherwise}, \end{cases} \tag{6}$$

where $\mathcal{B}(\boldsymbol{Q}_i, \boldsymbol{K})$ is the bias function for $\boldsymbol{Q}_i$, and $[x, y]$ represents the indices of the query and key. For $\boldsymbol{Q}_i$, a positive bias is applied to $e_i$, while a negative bias is applied to other events. The remaining text is treated as prompt context, with no bias applied.

Finally, we obtain the bias term for each segmented video query in a similar manner and concatenate them together to obtain the complete bias function $\mathcal{B}(\boldsymbol{Q}, \boldsymbol{K})$:

$$\mathcal{B}(\boldsymbol{Q}, \boldsymbol{K}) = \mathcal{B}(\boldsymbol{Q}_1, \boldsymbol{K}) \oplus \mathcal{B}(\boldsymbol{Q}_2, \boldsymbol{K}) \ldots \oplus \mathcal{B}(\boldsymbol{Q}_m, \boldsymbol{K}) \in \mathbb{R}^{N \times M}, \tag{7}$$

where $\oplus$ indicates the concatenation function.

**Attention Reinforcement.** We observe that when the attention between $\boldsymbol{Q}_i$ and $\boldsymbol{e}_i$ is not salient enough and the overall distribution is overly flat, it is still difficult to achieve temporal alignment solely through attention rearrangement. To address this, we further leverage attention reinforcement to adaptively strengthen the focus on the temporally aligned event by additionally introducing a reinforcement factor term $\mathcal{R}(\boldsymbol{Q}, \boldsymbol{K})$ to attention rearrangement.

Specifically, we first obtain the original distribution of attention probes $\boldsymbol{p}_i = \text{Softmax}\left(\frac{\boldsymbol{Q}_i \boldsymbol{K}^\top}{\sqrt{d}}\right)$, and measure the attention intensity of each text token after normalization as $\boldsymbol{p}_i' = \frac{\boldsymbol{p}_i - \boldsymbol{p}_i^{\min}}{\boldsymbol{p}_i^{\max} - \boldsymbol{p}_i^{\min} + \epsilon}$. Subsequently, we can adaptively adjust the positive strengthening factor $\boldsymbol{r}_i^+$ and the negative strengthening factor $\boldsymbol{r}_i^-$ based on $\boldsymbol{p}_i$. Specifically, when $\boldsymbol{p}_i'$ is small for a temporally aligned event $\boldsymbol{e}_i$ or large for other events, the intensity needs to be increased accordingly:

$$\boldsymbol{r}_i^+ = \boldsymbol{r}^{\min} + (1 - \boldsymbol{p}_i') \cdot (\boldsymbol{r}^{\max} - \boldsymbol{r}^{\min}), \tag{8}$$

$$\boldsymbol{r}_i^- = \boldsymbol{r}^{\min} + \boldsymbol{p}_i' \cdot (\boldsymbol{r}^{\max} - \boldsymbol{r}^{\min}), \tag{9}$$

where $\boldsymbol{r}^{\min}$ and $\boldsymbol{r}^{\max}$ are the lower and upper bounds of the strengthening factor, which are experimentally set to 1 and 1.5, respectively. Then $\mathcal{R}(\boldsymbol{Q}_i, \boldsymbol{K})$ can be formulated as:

$$\mathcal{R}(\boldsymbol{Q}_i, \boldsymbol{K})[x, y] = \begin{cases} \boldsymbol{r}_i^+[x, y], & \text{if } y \in e_i, \\ \boldsymbol{r}_i^-[x, y], & \text{if } y \in e_j, i \neq j \\ 0, & \text{otherwise}, \end{cases} \tag{10}$$

Finally, we obtain the complete $\mathcal{R}(\boldsymbol{Q}, \boldsymbol{K})$ to match $\mathcal{B}(\boldsymbol{Q}, \boldsymbol{K})$.

$$\mathcal{R}(\boldsymbol{Q}, \boldsymbol{K}) = \mathcal{R}(\boldsymbol{Q}_1, \boldsymbol{K}) \oplus \mathcal{R}(\boldsymbol{Q}_2, \boldsymbol{K}) \ldots \oplus \mathcal{R}(\boldsymbol{Q}_m, \boldsymbol{K}) \in \mathbb{R}^{N \times M}, \tag{11}$$

For simplicity, we illustrate the process of TS-Attn using the prompt containing only a single subject. The details for handling prompts with multiple subjects can be found in Appendix A.

# 4 EXPERIMENTS

## 4.1 EXPERIMENTAL SETUP

**Implementation Details.** We seamlessly integrate TS-Attn into multiple video generation models, including: (1) CogVideoX Yang et al. (2024) based on the MM-DiT architecture, and (2) Wan2.1

and Wan2.2 models Wang et al. (2025a) based on the Cross-DiT architecture, which injects text conditions through cross-attention. We perform both T2V and I2V tasks on these models. For the T2V task, TS-Attn is applied to the first 20% of the denoising steps. For the I2V task, the first 40% of the denoising steps are selected to enhance control effects. Basic inference settings such as the number of denoising steps, the scheduler type, and video resolution remain consistent with the original configurations of these models. All experiments are conducted on NVIDIA A100 GPU.

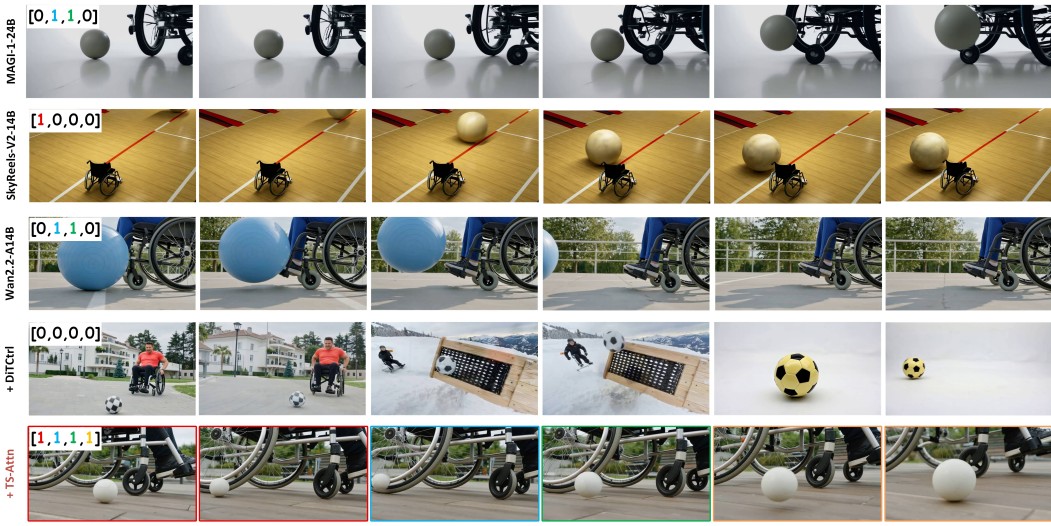

A ball rolls towards a wheelchair, collides the wheelchair and bounces back, and then rolls away slowly.

Figure 4: **Qualitative comparison results on multi-event T2V generation.** The list in the top-left corner, evaluated jointly by GPT-4o and humans, indicates the completion status of events.

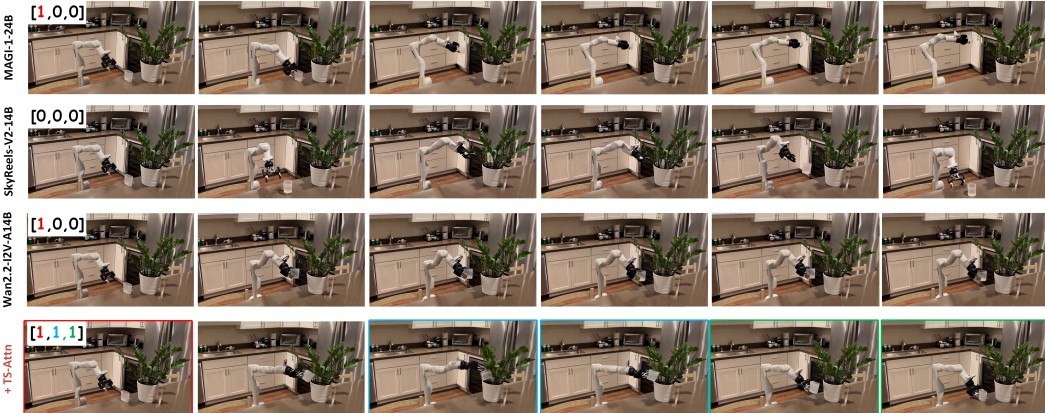

The robotic arm grasps the cup carefully, pours water into the potted plant, and then places the cup back on the counter.

Figure 5: **Qualitative comparison results on multi-event I2V generation.** The list in the top-left corner, evaluated jointly by GPT-4o and humans, indicates the completion rates. SkyReels-V2-14B generates actions that defy the laws of physics, resulting in a completion score of zero for all events.

**Baseline Models.** The comparison models we selected can be divided into three categories: (1) Basic video generation models, which include Open-Sora-Plan 1.3.0 Lin et al. (2024), Open-Sora 1.2 Zheng et al. (2024), Vchitect-2.0 Fan et al. (2025), Pyramid-Flow Jin et al. (2024), SkyReels-V2 Chen et al. (2025), and MAGI-1 Teng et al. (2025); (2) Multi-event video generation models, which includes MEVG Oh et al. (2024) and DiTCtrl Cai et al. (2025) reimplemented on Wan2.2-A14B; (3) Closed-sourced models, including KlingAI (2024), and HailuoAI (2024). Training-based methods MinT Wu et al. (2025b) and ShotAdapter Kara et al. (2025) are excluded due to closed-source code and data.

**Benchmark and Evaluation Metrics.** We select StoryEval-Bench Wang et al. (2025b) for the quantitative evaluation of multi-event T2V tasks, as it is a representative benchmark containing 423 prompts across seven classes, with 2–4 events per prompt. This benchmark utilizes GPT-4o OpenAI (2024) and LLaVA-OV-Chat-72B Li et al. (2024) to evaluate event completeness, temporal accuracy, and subject consistency in the generated videos. Since no existing multi-event I2V benchmark is available, we construct StoryEval-Bench-I2V. Specifically, GPT-4o is used to reparse each prompt to describe the initial state of the video, and Qwen-Image Wu et al. (2025a) synthesizes the first frame according to the reparsed prompt. Further details can be found in the Appendix B.

Table 1: **Evaluation results on T2V tasks with GPT-4o verifier.** Best scores are **bolded**.

| Model | Human | Animal | Object | Retrieval | Creative | Easy | Hard | Average |
|---|---|---|---|---|---|---|---|---|
| Hailuo | 38.2% | 38.3% | 27.5% | 42.6% | 18.0% | 58.9% | 9.7% | 35.1% |
| Kling-1.5 | 37.2% | 44.9% | 36.6% | 39.4% | 36.0% | 60.8% | 16.4% | 40.1% |
| Open-Sora-Plan 1.3.0 | 9.1% | 9.7% | 9.4% | 13.2% | 7.1% | 18.2% | 3.2% | 9.4% |
| Open-Sora 1.2 | 16.4% | 18.3% | 16.2% | 24.7% | 11.8% | 32.7% | 4.3% | 17.9% |
| Vchitect-2.0 | 21.5% | 19.9% | 20.4% | 22.0% | 15.2% | 42.8% | 3.9% | 21.7% |
| Pyramid-Flow | 17.8% | 16.5% | 12.8% | 23.4% | 9.7% | 35.1% | 1.0% | 16.0% |
| SkyReels-V2 | 43.8% | 39.9% | 35.4% | 43.1% | 27.0% | 55.9% | 26.7% | 40.6% |
| MAGI-1-24B | 39.6% | 32.7% | 33.5% | 41.9% | 24.8% | 51.7% | 20.5% | 35.8% |
| MEVG + Wan2.2-A14B | 47.7% | 39.7% | 40.5% | 47.6% | 28.3% | 57.8% | 28.9% | 43.1% |
| DiTCtrl + Wan2.2-A14B | 50.5% | 48.4% | 39.8% | 57.9% | 26.2% | 60.1% | 33.4% | 46.5% |
| CogVideoX-5B | 17.1% | 16.4% | 14.0% | 16.0% | 7.4% | 35.4% | 4.6% | 16.4% |
| **+Ours** | 28.0% | 25.4% | 21.7% | 32.9% | 13.9% | 45.7% | 9.9% | 25.8% |
| Wan2.1-1.3B | 32.4% | 31.0% | 24.9% | 30.6% | 22.1% | 42.3% | 17.6% | 29.1% |
| **+Ours** | 43.1% | 33.9% | 34.6% | 47.0% | 24.5% | 53.2% | 23.5% | 37.6% |
| Wan2.1-14B | 41.4% | 37.2% | 31.9% | 45.2% | 21.9% | 53.8% | 24.6% | 37.6% |
| **+Ours** | 54.7% | 50.0% | 45.1% | 62.1% | 35.2% | 64.5% | 38.7% | 50.2% |
| Wan2.2-A14B | 51.2% | 46.7% | 44.9% | 54.8% | 34.8% | 60.3% | 34.0% | 48.3% |
| **+Ours** | **60.4%** | **53.6%** | **52.0%** | **63.0%** | **45.3%** | **70.5%** | **44.3%** | **56.2%** |

Table 2: **Quantitative comparison results on I2V evaluation tasks with GPT-4o verifier.**

| Model | Human | Animal | Object | Retrieval | Creative | Easy | Hard | Average |
|---|---|---|---|---|---|---|---|---|
| Framepack-13B | 37.3% | 30.9% | 28.2% | 45.0% | 21.1% | 43.9% | 25.3% | 33.5% |
| SkyReels-V2-I2V-14B | 40.5% | 37.9% | 34.1% | 41.1% | 25.5% | 43.7% | 28.0% | 36.9% |
| MAGI-1-I2V-24B | 37.2% | 31.3% | 32.6% | 37.0% | 19.4% | 44.7% | 26.7% | 34.2% |
| CogVideoX-I2V-5B | 21.0% | 18.8% | 17.5% | 23.3% | 10.0% | 35.8% | 9.9% | 19.6% |
| **+Ours** | 28.2% | 28.8% | 23.5% | 35.1% | 16.5% | 44.3% | 15.9% | 28.3% |
| Wan2.1-I2V-14B | 43.8% | 33.9% | 36.0% | 42.1% | 29.8% | 44.4% | 31.9% | 37.0% |
| **+Ours** | 46.0% | 38.8% | 43.3% | 44.9% | 32.0% | 54.2% | 32.6% | 42.6% |
| Wan2.2-I2V-A14B | 48.4% | 49.3% | 43.1% | 50.3% | 34.4% | 57.8% | 39.1% | 47.5% |
| **+Ours** | **58.3%** | **53.2%** | **50.4%** | **63.0%** | **36.5%** | **64.0%** | **43.8%** | **54.4%** |

## 4.2 QUALITATIVE COMPARISON

Figure 1(a) presents representative examples generated by our method, showcasing its robust capability to handle multi-event generation tasks. In particular, Figure 4 illustrates results for multi-event T2V generation, where the ball interacts with a wheelchair, demonstrating a smooth sequence of events, including rolling, collision, and subsequent movement. Additionally, Figure 5 highlights multi-event I2V generation, showing a robotic arm performing tasks such as grasping, pouring, and placing an object. In both cases, our method effectively captures the interactions and transitions between events, with GPT-4o and human evaluations jointly assessing the completion status. This comparison underscores the model's ability to handle complex, multi-step sequences across various scenarios, emphasizing its effectiveness, and robust generalization in diverse video generation tasks.

## 4.3 QUANTITATIVE COMPARISON

**Benchmark Comparison.** As shown in Table 1, incorporating TS-Attn into Wan2.2-A14B, Wan2.1-14B, Wan2.1-1.3B, and CogVideoX-5B significantly improves baseline performance across different architectures and scales. For example, we observe relative improvements of 33.5% and

57.3% on the StoryEval-Bench score in Wan2.1-T2V-14B and CogVideoX-5B models, respectively. This clearly demonstrates the versatility of TS-Attn across various model architectures. Besides, when using Wan2.2-A14B as the baseline, TS-Attn significantly outperforms DiTCtrl and MEVG, which are based on the multi-prompt paradigm. This further demonstrates TS-Attn's excellent trade-off between temporal consistency and prompt-following.

Table 3: **Inference time comparison on a single A100 GPU for different models.**

| Model | SkyReels-v2-14B | MAGI-1-24B | Wan2.2-A14B | +MEVG | +DiTCtrl | +TS-Attn(Ours) |
|---|---|---|---|---|---|---|
| **Latency (s)** | 1865 | 2732 | 846 | 2453 | 2749 | 863 |

Table 4: **Ablation results of TS-Attn on StoryEval-Bench.**

| Method | Wan2.2-A14B | | | Wan2.1-14B | | | CogVideoX-5B | | |
|---|---|---|---|---|---|---|---|---|---|
| | **Easy** | **Hard** | **Avg** | **Easy** | **Hard** | **Avg** | **Easy** | **Hard** | **Avg** |
| **Baseline** | 60.3% | 34.0% | 48.3% | 53.8% | 24.6% | 37.6% | 35.4% | 4.6% | 16.4% |
| **+ EAM** | 66.2% | 39.8% | 51.9% | 62.6% | 31.1% | 46.4% | 42.1% | 7.3% | 22.9% |
| **+ EAM & MRE** | 70.5% | 44.3% | 56.2% | 64.5% | 38.7% | 50.2% | 45.7% | 9.9% | 25.8% |

In the I2V task, TS-Attn consistently brings performance improvements across various baseline models, as shown in Table 2. Overall, the extensive experiments above demonstrate the excellent performance of TS-Attn across various tasks and model architectures. Further quantitative comparisons evaluated using LLaVA-OV-Chat-72B are provided in the Appendix Table 5 and Table 6.

**Inference Efficiency Analysis.** We compare TS-Attn with other models in generating 480×832 videos to evaluate inference efficiency. For single-prompt models, the frame count is fixed at 81, while for multi-prompt models (e.g., DiTCtrl, MEVG), it is approximately $81 \times n$, where $n$ denotes the number of events in the prompt. The average response time for temporal segmentation using GPT-4o-mini is 2.65 seconds, which is also included in the overall inference time. The average inference time on StoryEval-Bench is used for comparison. As shown in Table 3, TS-Attn increases inference time by only 2% compared to Wan2.2-A14B, while significantly outperforming models like DiTCtrl and MAGI-1-24B.

## 4.4 Ablation Study

**Event-aware Attention Modulation.** We verify the effectiveness of event-aware attention modulation (EAM). As shown in Table 4, EAM significantly improves baseline performance by 23.4% on Wan2.1-14B and 39.6% on CogVideoX-5B, validating its effectiveness. We also conduct an in-depth analysis of the attention rearrangement and attention reinforcement subcomponents within EAM. As illustrated in Appendix Table 7, attention rearrangement contributes more to performance improvement, validating its effectiveness in temporally aligning multiple events. Attention reinforcement, on the other hand, serves more as a supporting component, adaptively adjusting the strength of attention rearrangement to accommodate diverse cases.

**Motion Region Extraction.** We also analyze the role of the Motion Region Extraction (MRE) module. As shown in Figure 7, MRE constrains attention modulation to motion-related regions, ensuring the precision of modulation while avoiding interference with the cross-attention distribution of background video tokens, thus preventing issues such as abrupt scene changes. Table 4 quantitatively validates the effectiveness of MRE.

**Different Temporal Segmentation Methods.** Finally, we discuss the impact of different temporal segmentation methods on performance. As shown in Appendix Table 8, the performance differences among uniform segmentation, human annotation, and GPT-4o-mini planning are minimal. This indicates that TS-Attn only requires a rough temporal segmentation to effectively perform reasonable attention reallocation. More discussions can be found in the Appendix E.

## 5 Conclusion

In this work, we introduce Temporal-wise Separable Attention (TS-Attn), a novel attention mechanism designed to address the challenges of generating videos from complex temporal descriptions.

The mechanism dynamically reallocates attention to ensure both temporal consistency and global coherence, effectively overcoming the trade-offs between action fidelity and prompt adherence observed in existing methods. Experimental results demonstrate that TS-Attn enhances the performance of pre-trained text-to-video models, yielding significant improvements in StoryEval-Bench scores with minimal impact on inference time. Moreover, TS-Attn operates as a plug-and-play solution, making it compatible with a variety of models for multi-event image-to-video tasks. This approach represents a significant advancement in scalable, high-quality video generation capable of handling complex and temporally dynamic input prompts.

ACKNOWLEDGMENTS

We thank all the anonymous reviewers for their constructive comments. This work was supported in part by the Peng Cheng Laboratory (No.2024KF1A00201).

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

# TS-Attn: Temporal-wise Separable Attention for Multi-Event Video Generation

## Appendix

## A  TS-ATTN FOR MULTIPLE SUBJECTS

For brevity of description, we introduce TS-Attn in the main text using a single subject and its corresponding event list. Therefore, this section provides a supplementary explanation for scenarios involving multiple subjects. Given a prompt with a subject list $[s_1, s_2, \ldots, s_m]$, we iteratively extract the motion-related region mask for each subject, resulting in $[M_{s_1}, M_{s_2}, \ldots, M_{s_m}]$. Similarly, based on the temporal distribution of the event list corresponding to each subject, we derive the attention rearrangement terms $[\mathcal{B}_{s_1}(Q, K), \mathcal{B}_{s_2}(Q, K), \ldots, \mathcal{B}_{s_m}(Q, K)]$ and the attention reinforcement terms $[\mathcal{R}_{s_1}(Q, K), \mathcal{R}_{s_2}(Q, K), \ldots, \mathcal{R}_{s_m}(Q, K)]$ for every subject. We can then obtain the final modulated attention map by summing the bias terms of all subjects:

$$A = \mathrm{softmax}\left(\frac{QK^\top + \sum_{i=1}^m M_{s_i} \odot \mathcal{R}_{s_i}(Q, K) \odot \mathcal{B}_{si}(Q, K)}{\sqrt{d}}\right) \in \mathbb{R}^{N \times M}, \quad (12)$$

It is worth noting that for multiple subjects, our implementation avoids repeated computation of the attention matrix. Instead, we only sequentially index the attention values at required positions for each subject to construct the bias terms. As a result, the inference overhead for multiple subjects remains nearly identical to that of a single subject.

## B  CONSTRUCTION PIPELINE FOR STORYEVAL-BENCH-I2V

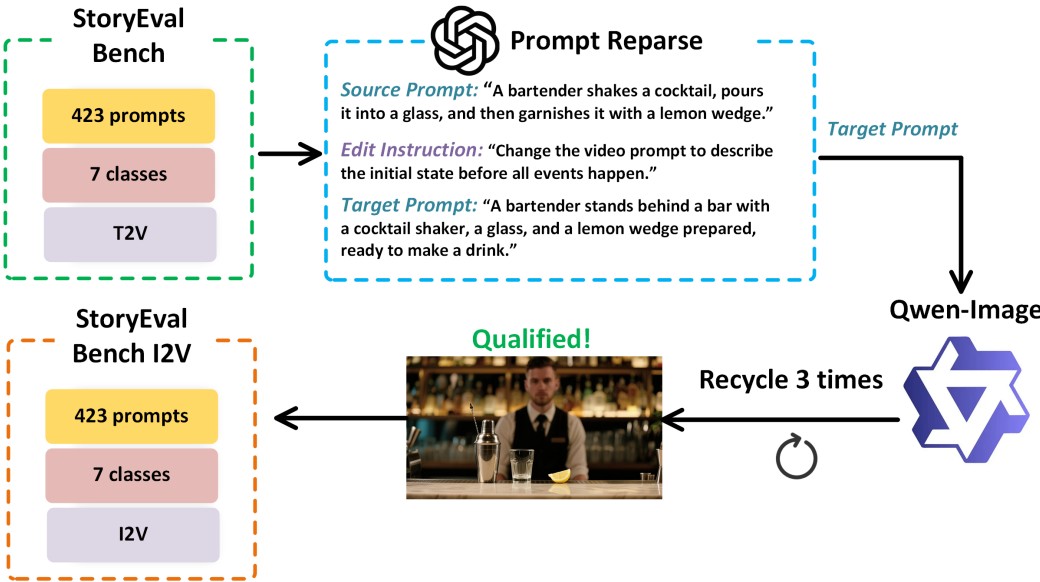

Figure 6: **Construction pipeline of StoryEval-Bench-I2V.**

Due to the absence of a dedicated multi-event I2V benchmark, we construct a new evaluation framework to assess the generalization ability of TS-Attn on I2V tasks. StoryEval-Bench Wang et al. (2025b), as a representative benchmark for multi-event text-to-video generation, has undergone peer review and features a large scale of prompts with high data diversity. Based on this foundation, we explore extending StoryEval-Bench to support the I2V task.

The core lies in deriving a reasonable first frame image from the video prompts in StoryEval-Bench. As illustrated in Figure 6, we first use GPT-4o OpenAI (2024) to convert source video prompts into

target descriptions of the initial state before any events occur. These descriptions primarily include static information such as the subjects and background layout involved in the video prompt, and can therefore be regarded as an approximate representation of the first frame of the video. We then employ the state-of-the-art text-to-image model Qwen-Image Wu et al. (2025a) to synthesize the first frame of the video based on the target descriptions. To ensure the accuracy of synthesized images, we select three different random seeds for synthesis and manually identify the optimal image. Through this process, we obtain 423 image-text pairs to support I2V task validation. Since we do not alter the prompt categories in StoryEval-Bench, we use the original benchmark's evaluation methodology for assessing the generated videos.

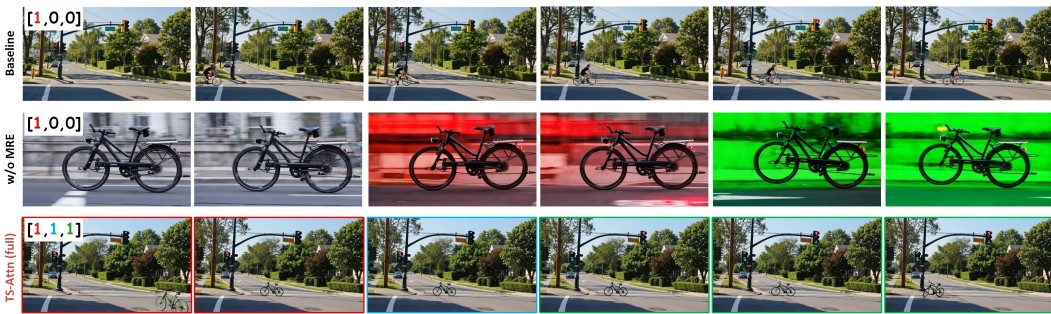

A bicycle pedals itself down the street, stops at a red light, and then continues when it turns green.

Figure 7: **Ablation results on the effect of motion region mask.** Not restricting attention modulation to motion-related regions can, in some cases, lead to background flickering, which ultimately degrades the overall video quality. Additionally, it hinders the motion regions from effectively responding to individual events.

## C MORE COMPARISON RESULTS WITH LLAVA-OV-CHAT-72B VERIFIER

As shown in Tables 5 and 6, we also employ the LLaVA-OV-Chat-72B Li et al. (2024) verifier to evaluate the generated videos. Consistent with the conclusions drawn using the GPT-4o verifier, TS-Attn consistently and significantly improves baseline performance across multiple models and both I2V and T2V tasks.

## D ABLATION RESULTS OF EAM

The core of TS-Attn, event-aware attention modulation, primarily consists of two sub-modules: attention rearrangement and attention reinforcement. To understand their individual contributions to performance, we conduct a more detailed ablation study in Table 7. It can be observed that removing attention rearrangement leads to a significant performance drop, further demonstrating that the more critical aspect of TS-Attn is the temporal redistribution of cross-attention distributions. Relying solely on attention reinforcement reduces TS-Attn to a mere attention enhancement mechanism for event tokens, lacking temporal correspondence. Combining both modules enables intensity-adaptive attention allocation and achieves optimal performance.

## E COMPARISON OF DIFFERENT TEMPORAL SEGMENTATION METHODS

We compare different temporal segmentation strategies that can be employed in TS-Attn.

**Uniform Segmentation.** This represents the simplest method for temporal segmentation: based on the number of events in the prompt, the video tokens are evenly divided into a corresponding number of intervals. In this setup, multiple events in the prompt are parsed by GPT-4o-mini.

**User Input.** Users can customize the intervals for each event based on the event count. For example, for a prompt containing four events, the video tokens can be partitioned in a ratio of 20%, 20%,

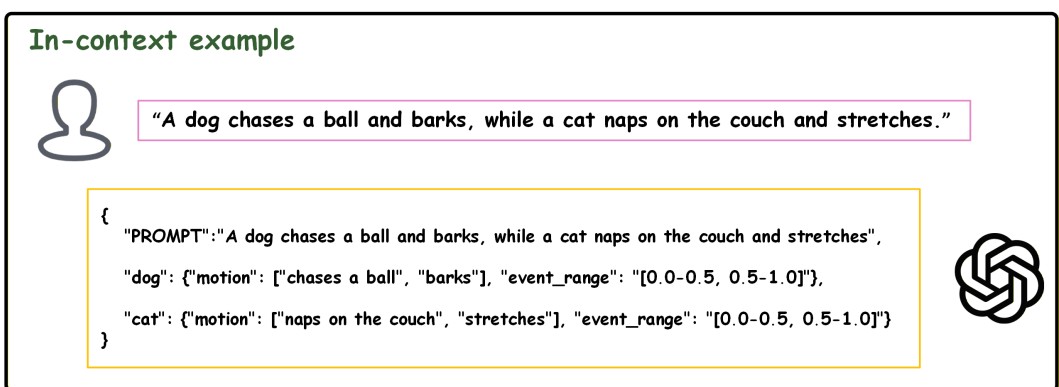

**System prompt**

You are a video analysis assistant. Your task is to divide the total video duration into time ranges corresponding to each event described in a given list of events. When performing this task, consider real-world physical constraints as well as the subject performing these events. The output should be a list of time ranges (as fractions of the total time) for each event, ensuring they sum to 1.0.

Specifically, given a prompt, you should first extract the subjects and motion components from it, then reasonably allocate intervals based on the order of events. Finally, please return a JSON file that hierarchically organizes these events in a temporal sequence. Below is an example:

**In-context example**

"A dog chases a ball and barks, while a cat naps on the couch and stretches."

```
{
    "PROMPT":"A dog chases a ball and barks, while a cat naps on the couch and stretches",
    "dog": {"motion": ["chases a ball", "barks"], "event_range": "[0.0-0.5, 0.5-1.0]"},
    "cat": {"motion": ["naps on the couch", "stretches"], "event_range": "[0.0-0.5, 0.5-1.0]"}
}
```

Figure 8: **The prompt template for temporal segmentation using the LLM API.**

30%, and 30% to align with each event. In the experiments summarized in Table 8, we manually annotated temporal intervals for each prompt in StoryEval-Bench based on commonsense knowledge.

**Efficient Planning with LLM API.** This approach is similar to user input: we instruct the LLM to partition reasonable temporal intervals for each prompt. Specifically, we employ the GPT-4o-mini for this segmentation task due to its simplicity. The LLM API processes each prompt in approximately 2 to 3 seconds, demonstrating high efficiency.

All three methods mentioned above are straightforward and easy to implement. As demonstrated in Table 8, their differences in final performance are minimal. This further confirms that even with only coarse temporal interval guidance, TS-Attn is capable of achieving temporal-aware multi-event video generation. Moreover, overlapping intervals between different events do not significantly impact performance, as TS-Attn employs a soft attention redistribution mechanism. Video tokens within a specific temporal interval are guided to focus primarily on attention interactions corresponding to their assigned event, rather than being completely isolated from other events. The prompt template we used is shown in Figure 8.

## F   MORE ATTENTION VISUALIZATION RESULTS

We present additional attention analysis to further elaborate on the insights of TS-Attn. As shown in Figure 9, the attention distributions of different actions in TS-Attn are clearly separated along the temporal sequence. Meanwhile, each event exhibits a strong response to the motion regions of its corresponding frames. This significantly enhances the temporal awareness of the original cross-attention and, as expected, results in videos that respond accurately to all actions.

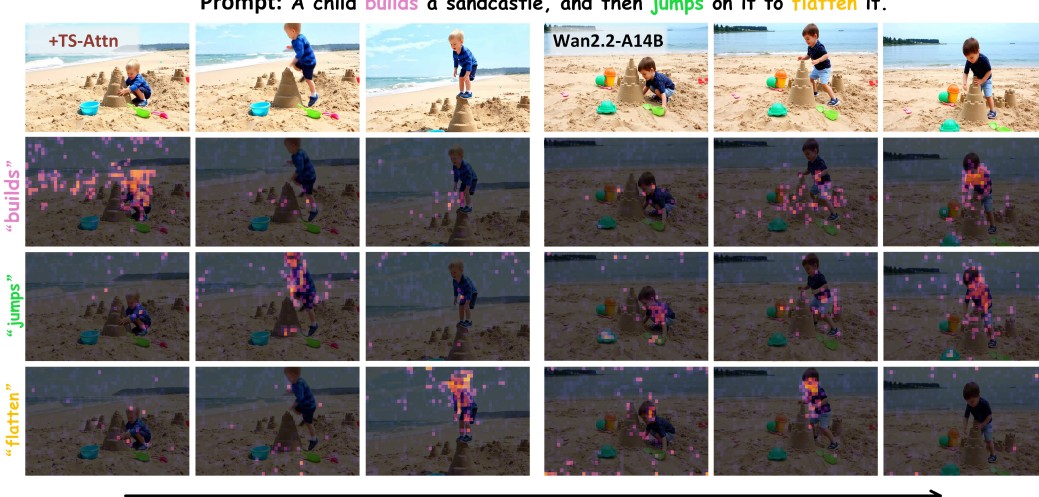

Figure 9: **More comparison of attention maps along the temporal sequence between TS-Attn and vanilla cross-attention.**

## G    MORE QUALITATIVE RESULTS

In this section, we provide additional qualitative comparisons to further demonstrate the effectiveness of our method on multi-event video generation tasks. Figures 10–15 present more text-to-video (T2V) cases under complex temporal prompts, where our approach consistently achieves coherent event transitions and maintains high visual fidelity. These results highlight the generalization ability of our model in handling diverse multi-event scenarios across different subjects and environments.

Moreover, Figures 16 and 17 showcase comparisons with Wan2.1-14B. Our method demonstrates stronger temporal consistency and better adherence to prompt semantics, especially in cases involving multiple interacting events. These results further validate the robustness and scalability of our approach beyond standard benchmarks.

## H    MORE COMPARISON WITH MULTI-PROMPT METHODS

VideoTetris Tian et al. (2024) and TALC Bansal et al. (2024) are frameworks that use multi-prompt strategies to address compositional generation and multi-scene generation, which share certain similarities with multi-event generation. To further expand our evaluation scope, we extend these frameworks to the multi-event generation task. Specifically, we implement VideoTetris and TALC on Wan2.2-A14B using the optimal hyperparameters specified in their original papers, ensuring a fair comparison with TS-Attn. As shown in Table 9, TS-Attn substantially outperforms both TALC and VideoTetris. TALC's strict conditioning of each segment on sub-prompts disrupts global coherence, leading to reduced performance. Although VideoTetris combines weighted global and local cross-attention, its lack of training distorts the original video latent distribution, resulting in quality degradation and minimal improvement. Qualitative visual comparisons are provided in Figure 18.

## I    MORE DIVERSE APPLICATIONS OF TS-ATTN

In this section, we present more potential application scenarios of TS-Attn, including multi-event generation involving multiple subjects, scene-level multi-event generation, and enhancing the potential for interactive long-video generation.

**Multi-subject multi-event generation.**    As shown in Figure 19, multi-event generation involving multiple subjects can be achieved by using attention rearrangement to dynamically bind each subject to its corresponding event in the temporal sequence while suppressing interference from other events.

In this way, TS-Attn greatly enhances the model's capability to handle complex spatial and temporal prompts.

**Scene-level multi-event generation.** Meanwhile, we also observe that TS-Attn can handle scene-level multi-event transitions, such as landscapes and video styles (Figure 20). It accurately interprets dynamic temporal changes, responds precisely to weather and artistic styles in each temporal segment, and smoothly completes the transitions.

**Interactive long video generation.** The Wan model typically supports generating videos of up to 5 seconds in length, which limits the number of events it can reasonably express to no more than 5. To handle more events, we applied TS-Attn to the recently proposed LongCat-Video-13.6B model Team et al. (2025), which natively supports video continuity. This enables us to distribute a larger number of events across multiple clips. For example, 9 events can be divided into 3 clips for generation while maintaining temporal consistency.

As illustrated in Figure 21, TS-Attn improves temporal awareness within each clip, greatly enhancing the capability to handle videos with a large number of events. The benefits of integrating TS-Attn into architectures like LongCat-Video are twofold: 1) For a fixed number of events, TS-Attn enables generation with fewer clips; 2) For a fixed number of clips, TS-Attn effectively manages more intricate temporal descriptions. These results highlight the potential of TS-Attn for both interactive and long-form video generation.

## J    THE USE OF LARGE LANGUAGE MODELS

We use large language models (LLMs) solely for polishing sentence structures and refining the language throughout the manuscript. All core aspects of this research, including central ideas, experimental design, data analysis, result interpretation, and conclusion derivation, are conducted entirely by the authors. The LLM serves only as an auxiliary tool and is not involved in any key aspects requiring academic judgment or creative intellectual input.

Table 5: **Evaluation results on T2V tasks with LLaVA-OV-Chat-72B verifier.** Best scores are **bolded**.

| Model | Human | Animal | Object | Retrieval | Creative | Easy | Hard | Average |
|---|---|---|---|---|---|---|---|---|
| *Closed-Source Model* | | | | | | | | |
| Hailuo | 48.0% | 40.1% | 35.6% | 51.7% | 19.5% | 58.3% | 17.1% | 41.0% |
| Kling-1.5 | 41.9% | 46.0% | 35.1% | 41.7% | 30.8% | 56.1% | 24.1% | 41.7% |
| *Open-Source Model* | | | | | | | | |
| Open-Sora-Plan 1.3.0 | 13.5% | 13.2% | 9.6% | 17.1% | 6.9% | 28.3% | 2.2% | 12.7% |
| Open-Sora 1.2 | 26.2% | 22.2% | 20.2% | 32.2% | 15.4% | 37.8% | 10.8% | 23.6% |
| Vchitect-2.0 | 33.4% | 30.5% | 33.6% | 33.6% | 20.5% | 51.4% | 19.1% | 31.6% |
| Pyramid-Flow | 23.6% | 20.0% | 15.8% | 26.4% | 10.5% | 38.1% | 4.5% | 20.3% |
| SkyReels-V2 | 47.6% | 47.2% | 40.6% | 56.9% | 33.2% | 60.5% | 30.1% | 45.9% |
| MAGI-1-24B | 45.4% | 38.8% | 38.6% | 48.7% | 25.2% | 55.8% | 26.2% | 41.2% |
| MEVG + Wan2.2-A14B | 55.4% | 46.2% | 45.2% | 55.8% | 33.9% | 58.7% | 35.0% | 48.5% |
| DiTCtrl + Wan2.2-A14B | 56.6% | 54.5% | 45.0% | 59.9% | 33.0% | 65.2% | 37.1% | 51.8% |
| CogVideoX-5B | 19.7% | 20.7% | 17.4% | 27.2% | 8.1% | 37.6% | 7.1% | 19.9% |
| **+Ours** | 32.4% | 29.8% | 25.7% | 39.9% | 18.5% | 48.2% | 15.8% | 29.6% |
| Wan2.1-1.3B | 37.6% | 37.7% | 27.1% | 33.6% | 21.5% | 45.6% | 26.4% | 34.4% |
| **+Ours** | 46.2% | 42.6% | 36.5% | 51.3% | 28.5% | 50.9% | 33.3% | 41.8% |
| Wan2.1-14B | 51.0% | 40.6% | 36.4% | 58.2% | 23.8% | 57.3% | 31.6% | 43.5% |
| **+Ours** | 60.4% | 55.9% | 50.6% | 67.6% | 40.3% | 73.7% | 41.3% | 55.9% |
| Wan2.2-A14B | 62.6% | 56.5% | 48.8% | 70.2% | 42.9% | 69.3% | 45.9% | 56.8% |
| **+Ours** | **70.6%** | **63.4%** | **58.0%** | **76.6%** | **48.9%** | **78.0%** | **50.2%** | **63.9%** |

Table 6: **Evaluation results on I2V tasks with LLaVA-OV-Chat-72B verifier.** Best/2nd best scores are **bolded**/underlined.

| Model | Human | Animal | Object | Retrieval | Creative | Easy | Hard | Average |
|---|---|---|---|---|---|---|---|---|
| Framepack-13B | 41.4% | 37.3% | 35.4% | 50.1% | 25.0% | 51.8% | 28.0% | 37.9% |
| SkyReels-V2-14B | 49.8% | 44.5% | 40.7% | 52.7% | 30.8% | 54.7% | 32.5% | 43.8% |
| MAGI-1-24B | 43.6% | 38.3% | 39.0% | 46.2% | 25.1% | 49.7% | 36.1% | 40.4% |
| CogVideoX-I2V-5B | 24.7% | 24.6% | 20.5% | 29.3% | 12.4% | 42.9% | 9.8% | 23.9% |
| **+Ours** | 37.9% | 36.4% | 30.8% | 43.0% | 20.2% | 52.2% | 22.1% | 35.3% |
| Wan2.1-I2V-14B | 48.6% | 38.4% | 36.0% | 53.2% | 26.7% | 53.0% | 28.2% | 41.4% |
| **+Ours** | 53.0% | 46.8% | 42.8% | 57.9% | 32.3% | 55.3% | 39.0% | 47.9% |
| Wan2.2-I2V-A14B | 56.7% | 51.0% | 47.6% | 61.6% | 35.3% | 62.2% | 41.3% | 52.0% |
| **+Ours** | **59.7%** | **58.8%** | **51.8%** | **66.8%** | **40.2%** | **67.5%** | **45.4%** | **57.6%** |

Table 7: **Ablation experiments on the subcomponents of Event-aware Attention Modulation.**

| Method | Wan2.2-A14B | | | CogVideoX-5B | | |
|---|---|---|---|---|---|---|
| | Easy | Hard | Avg | Easy | Hard | Avg |
| **w/o Attention Rearrangement** | 63.1% | 36.8% | 49.4% | 38.2% | 5.9% | 18.8% |
| **w/o Attention Reinforcement** | 67.4% | 41.2% | 53.5% | 41.8% | 8.4% | 23.6% |
| **TS-Attn** | 70.5% | 44.3% | 56.2% | 45.7% | 9.9% | 25.8% |

Table 8: **Ablation results of different temporal segmentation methods.**

| Method | Wan2.2-A14B | | | CogVideoX-5B | | |
|---|---|---|---|---|---|---|
| | **Easy** | **Hard** | **Avg** | **Easy** | **Hard** | **Avg** |
| Uniform Segmentation | 69.8% | 42.6% | 55.3% | 44.5% | 9.2% | 25.2% |
| User Input | 71.4% | 45.0% | 56.8% | 44.8% | 11.3% | 26.5% |
| GPT-4o-mini Plan | 70.5% | 44.3% | 56.2% | 45.7% | 9.9% | 25.8% |

Table 9: **More multi-event T2V comparison with multi-prompt methods using GPT-4o verifier.** Best scores are **bolded**.

| Model | Human | Animal | Object | Retrieval | Creative | Easy | Hard | Average |
|---|---|---|---|---|---|---|---|---|
| Wan2.2-T2V-A14B | 51.2% | 46.7% | 44.9% | 54.8% | 34.8% | 60.3% | 34.0% | 48.3% |
| + TALC | 50.9% | 45.4% | 44.1% | 56.2% | 33.8% | 60.6% | 31.9% | 47.1% |
| + VideoTetris | 53.0% | 46.5% | 46.8% | **63.6%** | 35.9% | 63.5% | 37.5% | 49.7% |
| **+ Ours** | **60.4%** | **53.6%** | **52.0%** | 63.0% | **45.3%** | **70.5%** | **44.3%** | **56.2%** |

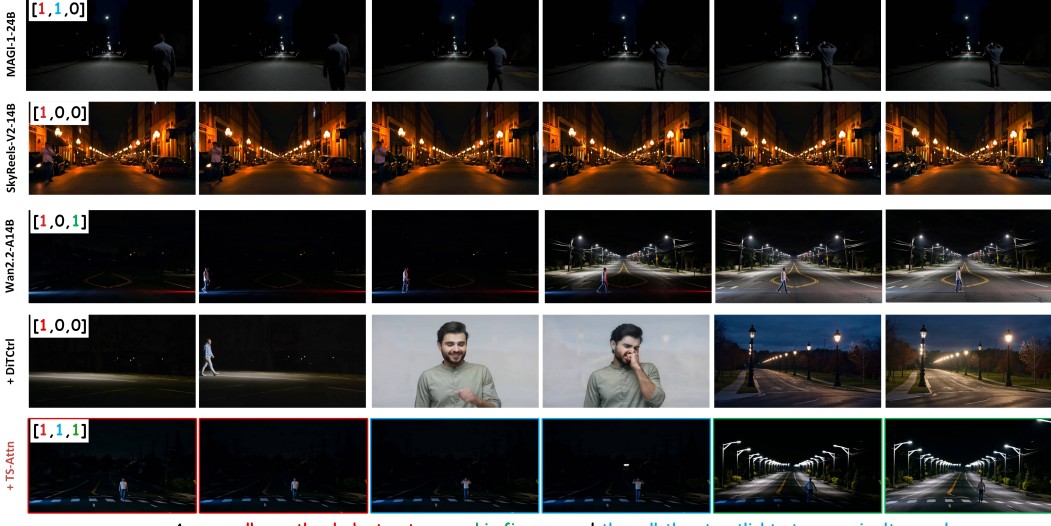

An elevator door opens, a dog goes out of the elevator, and then the door closes.

Figure 10: **More qualitative comparison results on multi-event generation.**

A man walks on the dark street, snaps his fingers, and then all the streetlights turn on simultaneously.

Figure 11: **More qualitative comparison results on multi-event generation.**

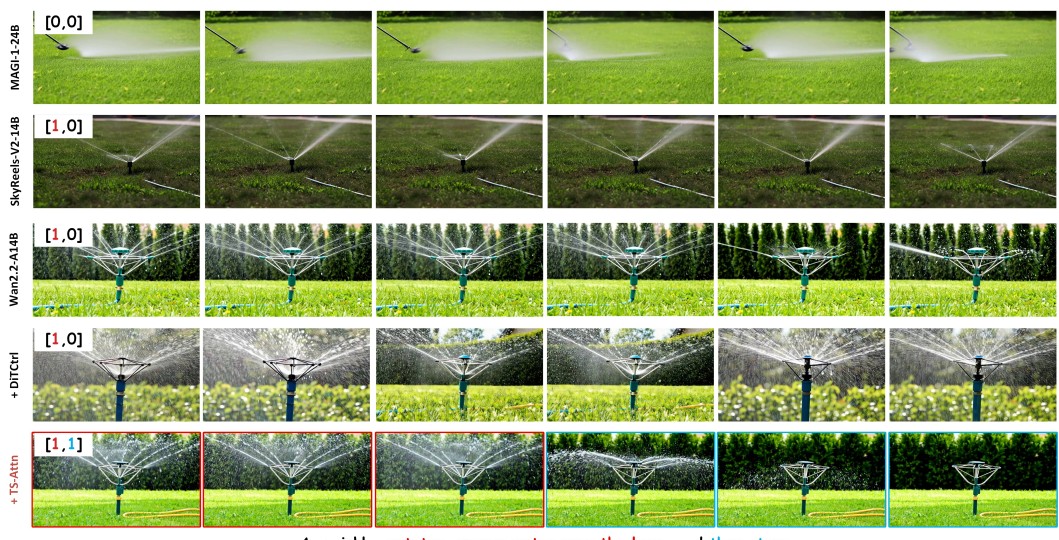

Figure 12: **More qualitative comparison results on multi-event generation.**

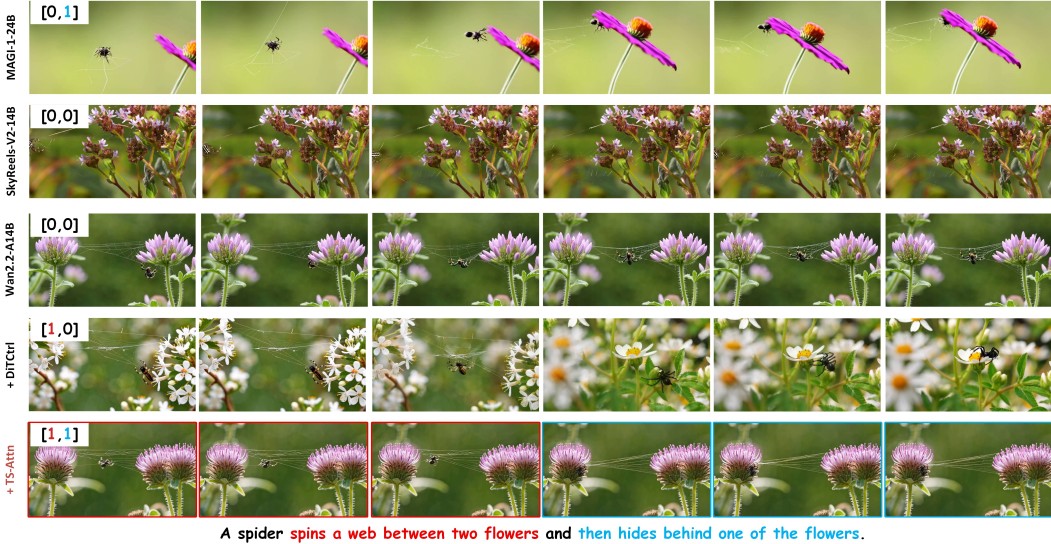

Figure 13: **More qualitative comparison results on multi-event generation.**

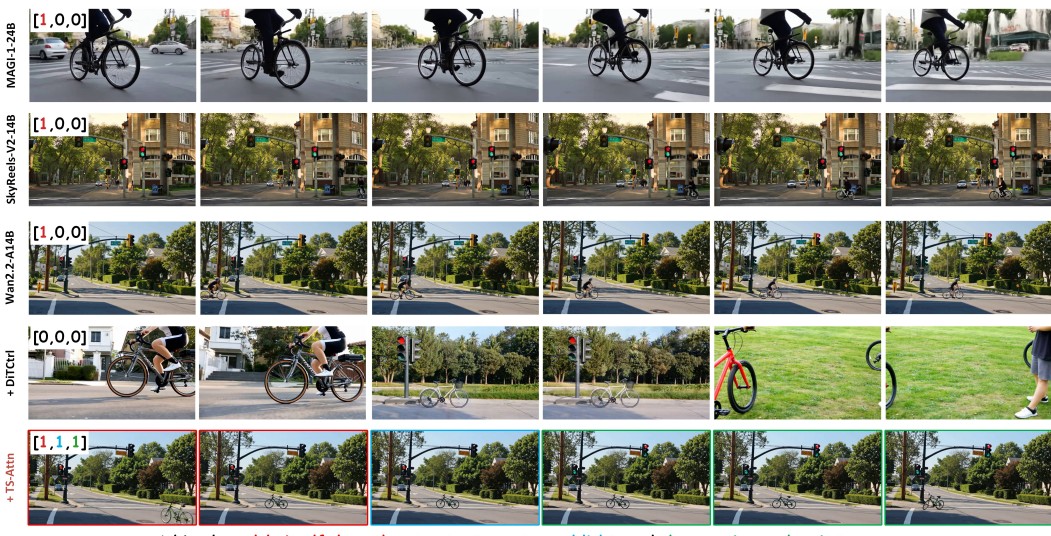

Figure 14: **More qualitative comparison results on multi-event generation.**

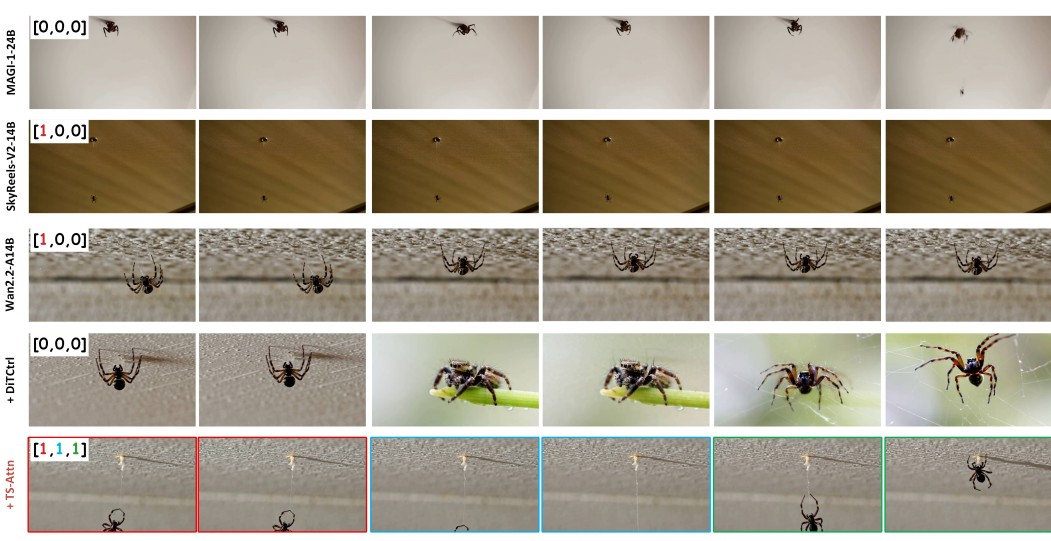

Figure 15: **More qualitative comparison results on multi-event generation.**

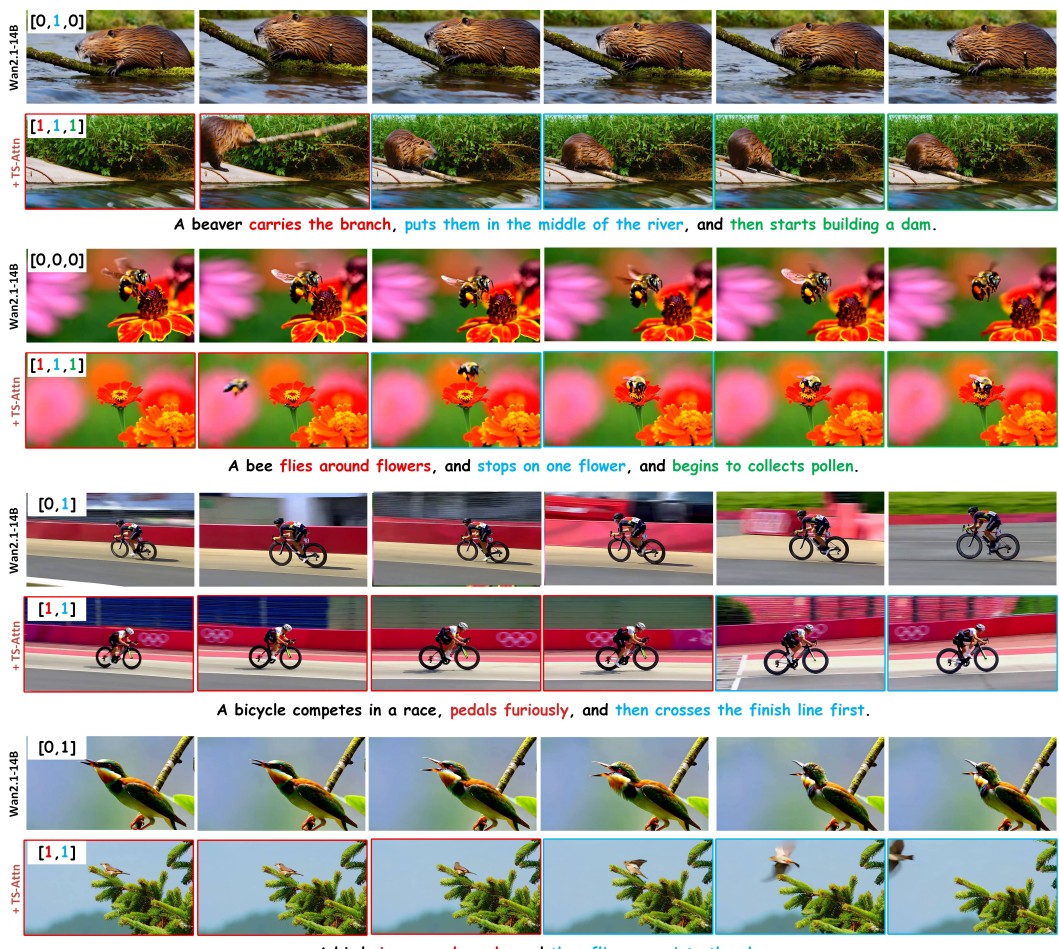

Figure 16: **More qualitative comparison results with Wan2.1-14B.**

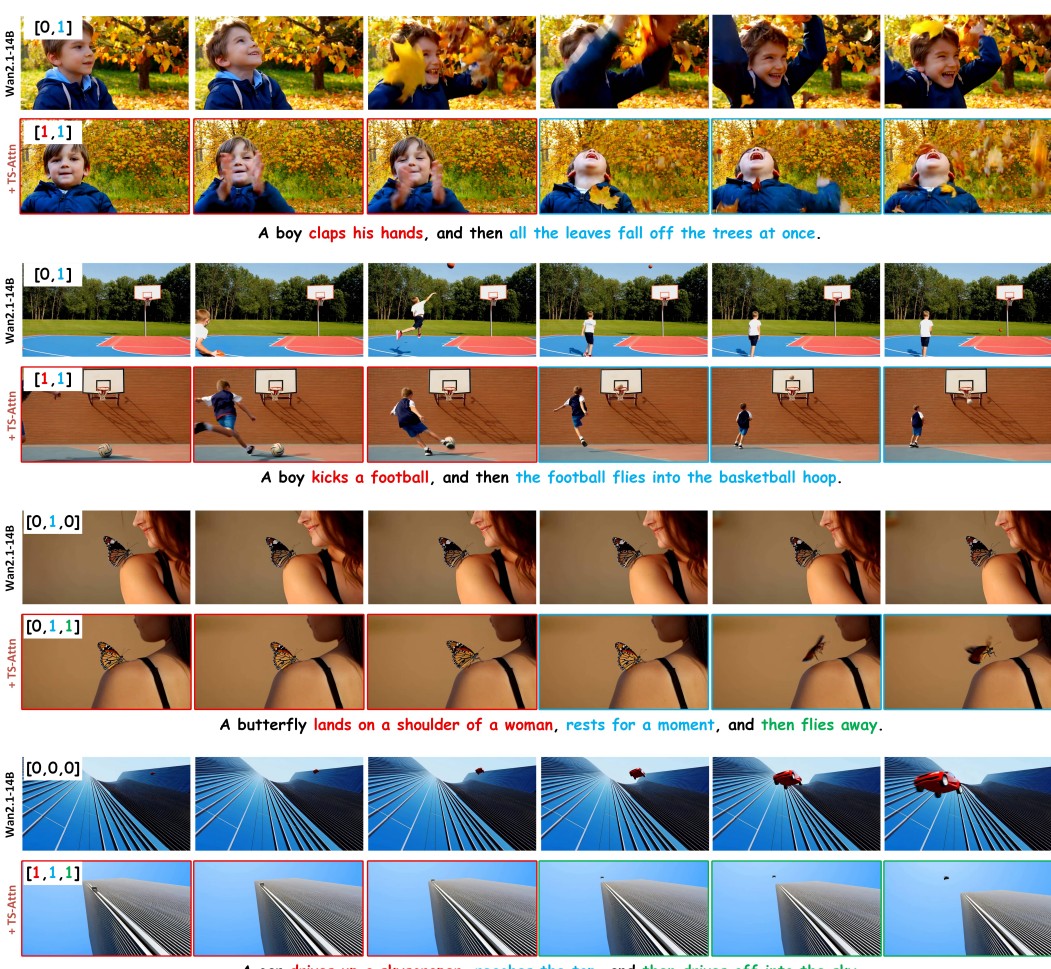

Figure 17: **More qualitative comparison results with Wan2.1-14B.**

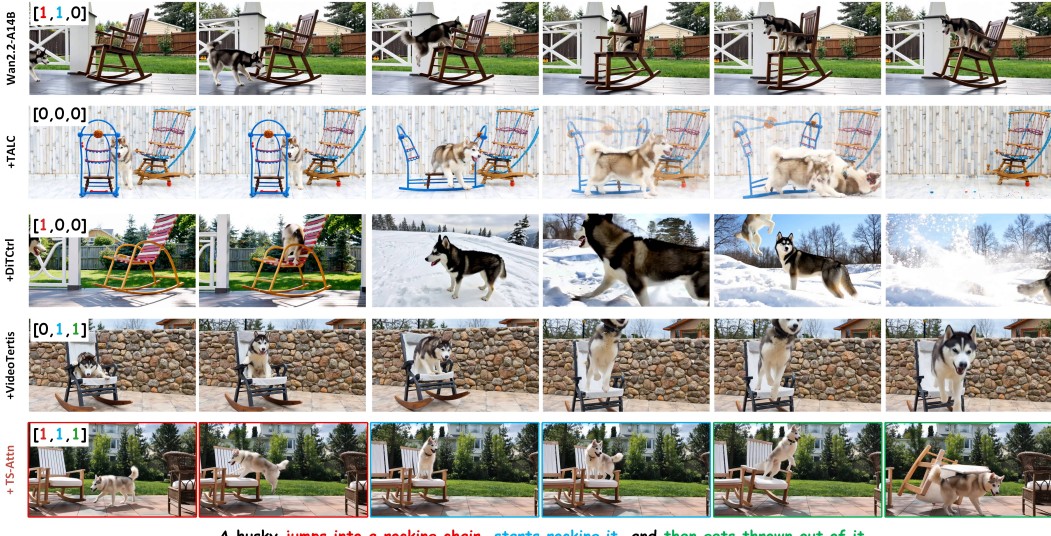

Figure 18: **More qualitative comparison results with multi-prompt methods.**

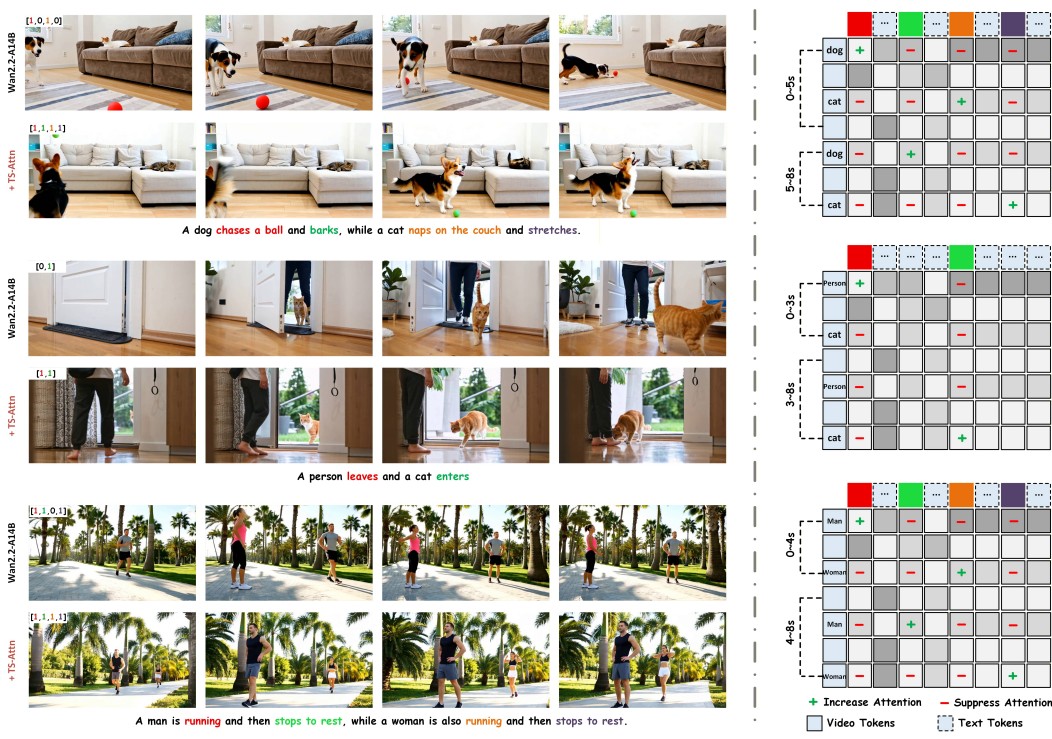

Figure 19: **More qualitative results on multi-event generation with multiple subjects.** The mask diagram on the right side of the figure briefly illustrates how attention rearrangement regulates the temporal attention intensity of each subject to different events under each prompt.

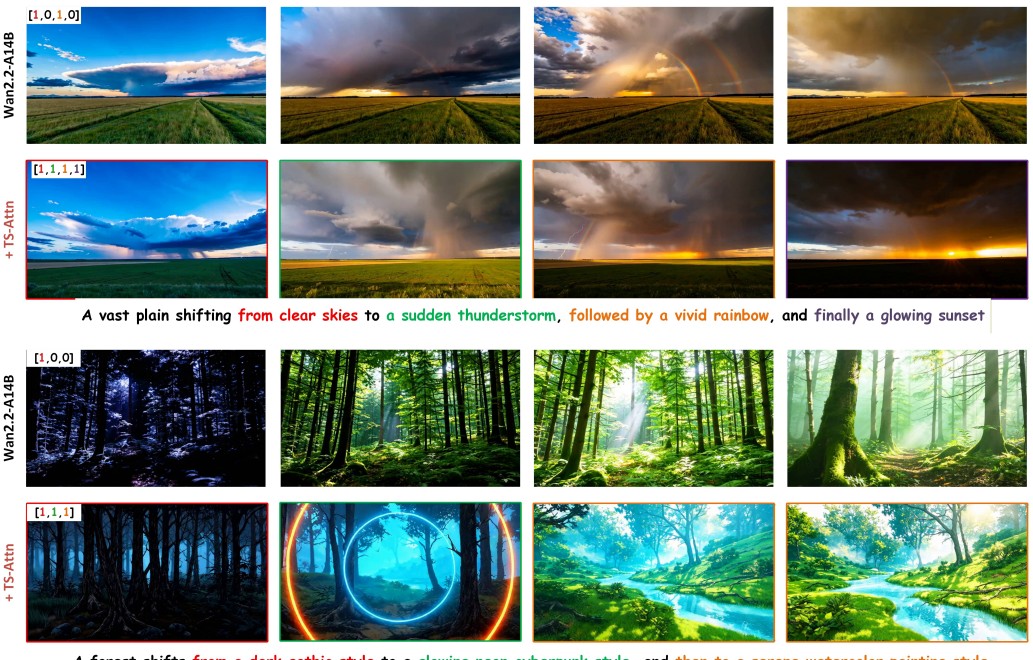

Figure 20: **More qualitative comparison results on scene-level multi-event generation.**

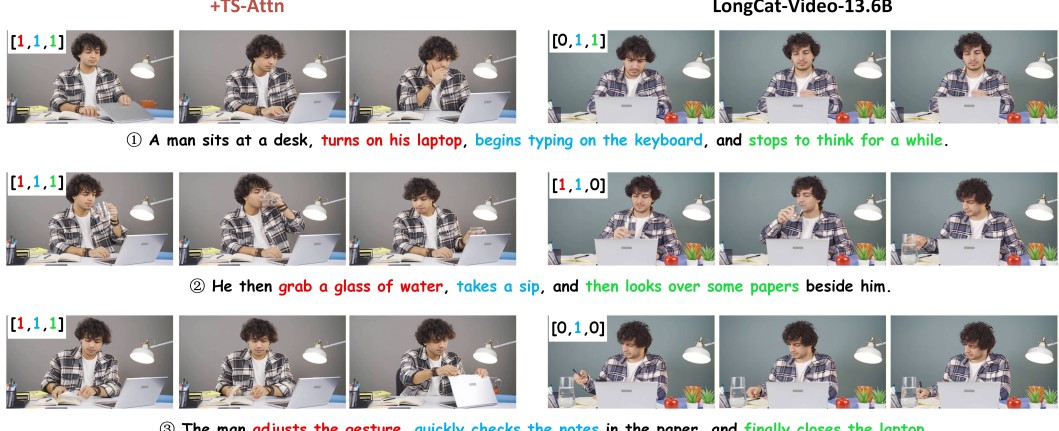

Figure 21: **More qualitative comparison results on interactive long video generation.**

