# OpenReview forum: "TS-Attn: Temporal-wise Separable Attention for Multi-Event Video Generation"
_ICLR.cc/2026/Conference — ICLR 2026 Poster_

### Official Review · Reviewer_oX7g · 2025-10-17

**Soundness:** 2
**Presentation:** 4
**Contribution:** 2
**Rating:** 6
**Confidence:** 4

**Summary:**

This work presents advancements in a typical issue presented by video generation methods, which often produce sequences of events that present temporal anomalies like overlapping events and incorrect ordering. This problem is also discussed in detail, to better understand its causes and justify the approach.

The solution presented is TS-Attn, a method that requires a modification to the cross-attention layer such that it makes use of event ordering information presented in the prompt itself. This expanded attention layer  requires additional input of temporal segmentation information, which is generated using either an external API, human data, or a simpler segmentation method.

Finally, the work does very complete experiments which (except for the caveats in the weaknesses section) appear to produce an important improvement to the provided baselines, while being a relatively simple to implement addition to the video generation systems.

**Strengths:**

The paper has substantial strengths:

1) the problem is worth solving, and the root cause analysis is excellent
2) the presented method is (to the extent of my knowledge) novel and interesting; event-aware attention modulation in particular seems like an interesting approach to me
3) the benchmark is very complete and required the integration to 3 different models, which is impressive
4) the benchmark results that are indicated in the paper are very strong, except for the points raised in the weaknesses section which will hopefully be easy to fix
5) finally, the paper is well written, and the explanation is very clear

**Weaknesses:**

This work presents a very complete analysis of an original method.

The soundness and contribution scores of this paper are diminished by ambiguity on the impact of latency in the full system, as explained below.

Specifically, the work would benefit from a more detailed explanation of how the uniform segmentation method works. When explaining this method, the paper refers to a number of events in the prompt, but it's unclear how the events in the prompt are parsed themselves. To me this seems like a critical point because if the event segmentation in the prompt has to be provided by a model for example, the time to run such segmentation should be accounted for in the latency analysis.

This applies generally to claims about latency, for example in figure 1c, and in other tables in the paper. It's unclear to me which segmentation method was used for each figure and table, whether segmentation itself was included in the latency analysis or not, and whether TS-Attn required additional human input that other methods didn't require (e.g. if uniform segmentation required manual selection of prompt events).

In summary, the paper could benefit from:

1) more clearly specifying how the uniform segmentation method works
2) providing details on which segmentation method was used for each figure and table
3) providing details on the impact of segmentation on latency

My understanding is that in most figures the LLM API approach is used, and it only adds about 2-3 seconds to latency (changing the +2% latency claim to 2.2% or 2.3% perhaps), but it would be good to be more explicit about this.

More minor points:

4) including the prompts used in the segmentation methods in the appendix could be an interesting addition.
5) that figure 2 makes use of 3 colors, 2 of which can be easily confused with each other.

**Questions:**

My main questions are about the segmentation pipeline and full impact on latency, which are discussed in detail in the weaknesses section.

---

> ### Author Response · Authors · 2025-11-24
> **Response to Reviewer oX7g (1/2)**
>
> We sincerely appreciate the reviewer's valuable suggestions and thoughtful reminders. We will first elaborate on the details of the temporal segmentation methods, followed by an introduction to the optimizations made in the paper presentation based on your suggestions.
>
> > ### **Weakness 1: More discussion on temporal segmentation technique**
>
> Thanks for your constructive suggestions. We will address these issues one by one to enhance the clarification of the paper.
>
> - **Default segmentation method used for each figure and table.**
> We use GPT-4o-mini for temporal segmentation and event extraction from the prompt, unless otherwise specified. Using LLM for simple prompt processing is highly efficient, allowing users to obtain structured inputs with just a single query in 2-3 seconds, which has been widely applied in previous video generation domains [1][2]. As a result, GPT-4o-mini is used by default for temporal planning in all our experiments. We have explicitly stated this in the revised **Section 3.2** to make the implementation details clearer.
>
> - **More details about uniform segmentation.**
> The list of events to be parsed in uniform segmentation is directly inherited from the list generated by GPT-4o-mini. Specifically, the interval planning for events originally provided by GPT-4o-mini is replaced with the even interval, thereby achieving uniform segmentation. For example, in a case with four events, we replace the GPT-4o-mini planned intervals of [0.2, 0.3, 0.3, 0.2] with uniform intervals of [0.25, 0.25, 0.25, 0.25] to achieve uniform segmentation.
> Notably, the comparison between uniform segmentation, user input, and GPT-4o-mini planned intervals is primarily introduced to demonstrate that TS-Attn only requires a coarse-grained temporal guide to achieve temporal-aware multi-event generation. We also present the ablation results of different temporal segmentation methods in **Table A** below. In practical use, we strongly recommend that users utilize LLMs for simple and efficient temporal planning. We have added additional explanations in **Appendix E** to make our descriptions more explicit.
>
> - **More details on the impact of segmentation on latency.**
> The inference time already includes the response time for temporal planning using GPT-4o-mini, with an average response time of 2.65 seconds. Thus, the total time is approximately 863 ≈ 2.65 (LLM API response) + 860 (TS-Attn inference), as shown in **Table B** below. We have clarified this point in the revised Section 4.3, part "Inference Efficiency Analysis." Thank you once again for your thoughtful reminder!
>
>
> **Table A: Ablation results of different temporal segmentation methods.**
>
> | Method               | Wan2.2-A14B Easy | Wan2.2-A14B Hard | Wan2.2-A14B Avg | CogVideoX-5B Easy | CogVideoX-5B Hard | CogVideoX-5B Avg |
> |----------------------|------------------|------------------|-----------------|-------------------|------------------|------------------|
> | Uniform Segmentation | 69.8%            | 42.6%            | 55.3%           | 44.5%             | 9.2%             | 25.2%            |
> | User Input           | 71.4%            | 45.0%            | 56.8%           | 44.8%             | 11.3%            | 26.5%            |
> | GPT-4o-mini Plan     | 70.5%            | 44.3%            | 56.2%           | 45.7%             | 9.9%             | 25.8%            |
>
>
>
> **Table B: Inference time comparison on a single A100 GPU for different models.**
>
> | Model                 | Latency (s) |
> |-----------------------|-------------|
> | SkyReels-v2-14B       | 1865        |
> | MAGI-1-24B            | 2732        |
> | Wan2.2-A14B           | 846         |
> | +MEVG                 | 2453        |
> | +DiTCtrl              | 2749        |
> | +TS-Attn(w/o LLM API) | 860         |
> | +TS-Attn(w/ LLM API)  | 863         |
>
>
> [1] Yuan, Shenghai, et al. "Identity-preserving text-to-video generation by frequency decomposition." CVPR. 2025.
>
> [2] Lian, Long, et al. "Llm-grounded video diffusion models." ICLR. 2024.

---

> ### Author Response · Authors · 2025-11-24
> **Response to Reviewer oX7g (2/2)**
>
> > ### **Weakness 2: Minor issues regarding paper presentation**
>
> We appreciate the reviewer's kind suggestions regarding the presentation of the paper. The specific details of our revisions are outlined as follows.
>
> - **Add prompt template used in the paper**
>
> We have added the prompt template for temporal planning in **Figure 8**. We will also provide detailed documentation after the code release to facilitate user usage!
>
> - **Adjust figure color**
>
> Thank you for your suggestion. We have adjusted the colors in **Figure 2** and **Figure 9** to enhance contrast and improve the presentation of the experimental results.

---

> > ### Comment · Reviewer_oX7g · 2025-11-25
> >
> > Thank you very much for your replies. Your comments satisfy my questions, especially those regarding latency concerns. I also appreciate your replies to other reviewers, especially those to reviewer Qohm, which handle other concerns I hadn't realized myself initially.
> >
> > Given this additional information provided, I'm raising my rating.

---

> > > ### Author Response · Authors · 2025-11-26
> > > **Response to Reviewer oX7g for Your Feedback**
> > >
> > > Dear Reviewer oX7g,
> > >
> > > We sincerely thank you for recognizing our work and offering valuable suggestions to improve it. We are pleased to have addressed your concerns and greatly appreciate your positive evaluation of the paper. Thank you for considering it favorably for acceptance.
> > >
> > > Once again, we truly appreciate the time and effort you dedicated to reviewing our work！

---

### Official Review · Reviewer_3mMY · 2025-11-01

**Soundness:** 3
**Presentation:** 3
**Contribution:** 3
**Rating:** 6
**Confidence:** 3

**Summary:**

This paper proposes Temporal-wise Separable Attention, a training-free method to improve video generation with complex multi-event prompts. The proposed attention mechanism enables this by dynamically restructuring cross-attention distributions to ensure motion-related regions in each frame primarily attend to temporally aligned events, demonstrating superior results across various video generation models.

**Strengths:**

- The problem is well-defined and the solution is intuitively designed. The visualization in Figure 2 makes this more convincing.
- The proposed Attention Rearrangement and Attention Reinforcement are carefully designed to better inject multi-prompts while maintaining pre-trained generative priors in a training-free manner.
- The experimental results are extensive. The proposed method has been implemented on various pre-trained models to demonstrate its effectiveness and is also compared with other methods for multi-prompts.

**Weaknesses:**

- There are some heuristics arising from the training-free design. For example, the erosion function and the accompanying spatial separation of subject tokens limit the applicable scenarios of this method to simple ones. For instance, cases where the subject is the style of the video or cannot be clearly distinguished in 2D spatial aspects fall outside this premise.
- While the attached video results look good, there is an absence of video comparisons with other multi-prompt methods. More qualitative comparisons in the paper would also be beneficial.

**Questions:**

Are there any issues with sudden scene changes?

---

> ### Author Response · Authors · 2025-11-24
> **Response to Reviewer 3mMY (1/3)**
>
> We sincerely thank the reviewer for the valuable suggestions and time. In our response, we first demonstrate the application of TS-Attn to broader scenarios, including multi-subject multi-event generation, large-scene multi-event generation, and style multi-event generation. Next, we supplement videos generated by multi-prompt methods and add comparison experiments with another two multi-prompt-based approaches. Finally, we present cases where TS-Attn handles sudden scene changes. The detailed content is as follows:
>
> > ### **Weakness 1: More discussion on motion region extraction**
>
> Thanks for your insightful question! The primary goal of motion region extraction is to ensure that, when handling subjects with relatively small foregrounds, each event can be accurately bound to the most relevant motion region in the temporal sequence. This is crucial for capturing fine-grained motion changes and maintaining temporal consistency.
>
> We also observe that motion region extraction can be extended to large-scale scenes or video style transitions, such as the following two prompts:
>
> - "A vast plain shifting from clear skies to a sudden thunderstorm, followed by a vivid rainbow, and finally a glowing sunset."
> - "A forest transitions from a dark gothic pencil sketch style to a glowing neon cyberpunk style, and then to a serene watercolor painting style."
>
> In this setup, the motion region mask for "plain" and "forest" will cover most of the frames, allowing these video tokens to participate in event-aware attention modulation. As validated in **Figure 20**, TS-Attn enhances the response to each event and ensures accurate temporal correspondence in such cases. This demonstrates the robustness of TS-Attn when extended to other forms of temporally complex prompts.

---

> ### Author Response · Authors · 2025-11-24
> **Response to Reviewer 3mMY (2/3)**
>
> > ### **Weakness 2: Multi-prompt videos & More qualitative comparisons**
>
> Thanks for your question! We have **uploaded the video demos corresponding to all the Figures** in the main text to the supplementary materials, including the multi-prompt method DiTCtrl [1]. We also extend the original TALC [2] and VideoTetris [3] frameworks, which are respectively based on multi-scene and compositional video generation, into temporal multi-prompt generation methods. Specifically, TALC divides the video into multiple clips, with each clip conditioned on its corresponding sub-prompt using hard-masked attention. For VideoTetris, we build on TALC by adding an additional cross-attention conditioning on the full prompt and combining the video latents updated by the sub-prompt and the full prompt through weighted addition.
>
> We implement TALC [2] and VideoTetris [3] on Wan2.2-A14B according to their original papers to ensure a fair comparison with TS-Attn. As shown in **Table A**, TS-Attn significantly outperforms both methods. TALC conditions each segment strictly on sub-prompts, which disrupts global coherence and reduces performance. VideoTetris incorporates weighted global and local cross-attention, but the lack of training distorts latent distributions, leading to quality degradation and limited improvement. Detailed qualitative comparisons can be found in **Figure 18**.
>
> **Table A: More multi-event T2V comparison with multi-prompt methods using GPT-4o verifier**
>
> | **Model**            | **Human** | **Animal** | **Object** | **Retrieval** | **Creative** | **Easy** | **Hard** | **Average** |
> |-----------------------|-----------|------------|------------|---------------|--------------|----------|----------|-------------|
> | Wan2.2-T2V-A14B      | 51.2%     | 46.7%      | 44.9%      | 54.8%         | 34.8%        | 60.3%    | 34.0%    | 48.3%       |
> | + TALC               | 50.9%     | 45.4%      | 44.1%      | 56.2%         | 33.8%        | 60.6%    | 31.9%    | 47.1%       |
> | + VideoTetris        | 53.0%     | 46.5%      | 46.8%      | **63.6%**     | 35.9%        | 63.5%    | 37.5%    | 49.7%       |
> | **+ Ours**           | **60.4%** | **53.6%**  | **52.0%**  | 63.0%         | **45.3%**    | **70.5%**| **44.3%**| **56.2%**   |
>
>
> We also provide additional qualitative results, showcasing the broader applicability of TS-Attn, including multi-subject multi-event generation (**Figure 19**), video-style multi-event generation (**Figure 20**), and interactive long video generation (**Figure 21**).
>
>
> [1] DiTCtrl: Exploring Attention Control in Multi-Modal Diffusion Transformer for Tuning-Free Multi-Prompt Longer Video Generation, 2025.
>
> [2] TALC: Time-Aligned Captions for Multi-Scene Text-to-Video Generation, 2024.
>
> [3] VideoTetris: Towards Compositional Text-to-Video Generation, 2024.

---

> ### Author Response · Authors · 2025-11-24
> **Response to Reviewer 3mMY (3/3)**
>
> > ### **Question 1: More discussion on sudden scene change**
>
> Thank you for your question. We will address it from two aspects: 1) How TS-Attn performs when the prompt includes sudden scene transitions; 2) Whether TS-Attn encounters temporal issues, such as sudden scene changes, in standard multi-event prompts. The details are described below：
>
> **Aspect 1: How TS-Attn performs when the prompt includes sudden scene transitions？**
>
> We have added qualitative results of TS-Attn handling multiple events with significant scene transitions. As shown in **Figure 20**, TS-Attn performs well even in examples where the scene changes abruptly over time. All events are accurately presented, and the scene transitions are relatively smooth.
>
>
> **Aspect 2: Whether TS-Attn encounters sudden scene changes in standard multi-event prompts?**
>
> We did not observe any significant sudden scene change issues, as confirmed by qualitative comparative experiments. This is mainly because TS-Attn adopts temporally soft and dynamic attention modulation rather than relying on hard-masked attention. Consequently, video tokens from different temporal segments can still interact with all events to some extent, ensuring temporal smoothness and global consistency.
>
> To quantitatively demonstrate this, we utilize four Temporal Quality metrics from VBench: subject consistency, background consistency, temporal flickering, and motion smoothness, to verify whether the incorporation of TS-Attn introduces any temporal issues, such as sudden scene changes. We evaluate these metrics on videos generated from all prompts in the StoryEval-Bench.
>
> As shown by the metrics in **Table B**, the addition of TS-Attn maintains background coherence and frame-to-frame stability comparable to the baseline, while providing slight improvements in motion smoothness and subject consistency. These improvements arise from TS-Attn’s ability to align motion regions in each frame with their corresponding events, which (1) ensures more natural action transitions and (2) reduces the risk of subject distortion during event changes. In summary, TS-Attn enhances overall temporal quality without introducing noticeable artifacts such as sudden scene changes or flickering.
>
> **Table B: More temporal quality comparison**
>
> | **Model**        | **Mot. Smoo.** | **Subj. Cons.** | **Back. Cons.** | **Temp. Flick.** | **Average** |
> |-------------------|----------------|------------------|------------------|------------------|-------------|
> | Wan2.2-A14B      | 0.9765         | 0.8762           | **0.9187**       | **0.9526**       | 0.9310      |
> | **+ Ours**       | **0.9814**     | **0.8821**       | 0.9158           | 0.9514           | **0.9327**  |

---

### Official Review · Reviewer_Qohm · 2025-11-01

**Soundness:** 3
**Presentation:** 3
**Contribution:** 2
**Rating:** 4
**Confidence:** 4

**Summary:**

This paper proposes TS-Attn, a training-free attention mechanism that improves multi-event video generation by dynamically separating and modulating cross-attention between motion regions and multi-event textual conditions. By introducing motion region extraction and event-aware attention modulation, the method reduces temporal misalignment and cross-event coupling, achieving better temporal coherence and event accuracy. TS-Attn can be plugged into existing diffusion-based video models without retraining, yielding substantial performance gains on StoryEval-Bench with only ~2% extra inference cost.

**Strengths:**

- Well-motivated and intuitive idea that directly targets temporal attention entanglement in multi-event video generation.

- Training-free and plug-and-play design makes it broadly applicable across existing diffusion models.

- Extensive experiments and ablations demonstrate consistent improvements and robustness across architectures and benchmarks.

- Clear presentation and visualizations that effectively explain both the mechanism and empirical benefits.

**Weaknesses:**

**Lack of comparison with prior methods**
The paper omits several highly relevant works that also manipulate cross-attention maps to achieve fine-grained event grounding without retraining, such as DreamRunner [1], VideoTetris [2], and TALC [3].
These methods similarly align textual tokens with corresponding visual regions through attention reweighting, making them conceptually close to TS-Attn. However, the authors neither cite nor compare with them. Including these approaches as baselines or at least discussing their differences would strengthen the paper’s positioning and contribution clarity.

**Unclear generalization to multi-subject scenarios**
The proposed motion-region extraction appears to assume a single dominant subject, computing masks for the entire video latents.
In cases involving subject transitions (e.g., a person leaves and a cat enters), this design may fail to isolate subject-specific motion regions, leading to incorrect or conflicting event-to-visual grounding.
Clarifying how TS-Attn handles such cases, or showing examples involving multiple subjects, would improve the completeness of the work.

---
[1] DreamRunner: Fine-Grained Compositional Story-to-Video Generation with Retrieval-Augmented Motion Adaptation, 2024.

[2] VideoTetris: Towards Compositional Text-to-Video Generation, 2024.

[3] TALC: Time-Aligned Captions for Multi-Scene Text-to-Video Generation, 2024.

**Questions:**

See weakness. Overall, I lean towards borderline for the current version and am happy to update my rating if my questions are well answered.

---

> ### Author Response · Authors · 2025-11-24
> **Response to Reviewer Qohm (1/2)**
>
> We appreciate the reviewer's constructive suggestions and the time spent reviewing our paper. First, we will use discussions and experiments to explain the differences in motivation and implementation between TS-Attn and the aforementioned three methods. Then, we will clarify how TS-Attn handles multiple subjects through more detailed descriptions and visualization cases.
>
> > ### **Weakness 1: More comparison with prior methods**
>
> Thanks for your question! We will include these methods in the related work section for further discussion. While these approaches share some relevance with TS-Attn, there are significant differences in both their motivation and implementation. DreamRunner and VideoTetris focus on multi-subject spatial relationships using hard-masked attention, unlike TS-Attn's soft temporal modulation. TALC targets multi-scene generation with isolated attention for temporal segments, which causes background consistency issues, making it less suited for multi-event tasks requiring both subject and background coherence.
>
> The detailed discussion and comparative experiments between TS-Attn and these methods are presented below：
>
> - **Discussion:**
>
>     - **Dreamrunner [1]**: It focuses on spatial control for multi-subject T2V generation, not temporal attention for multi-event generation. Methodologically, it uses hard-masked attention, while TS-Attn employs temporally-aware soft attention. Unlike TS-Attn, DreamRunner achieves multi-scene generation with subject consistency relying on personalized ID injection, aligning more with storytelling visualization.
>
>     - **VideoTertis [2]**: Videotetris focuses on compositional scene generation with multiple subjects, emphasizing interactions between spatial layouts and sub-prompts. While it mentions temporal prompt decomposition, its main goal is to ensure sub-prompt content appears in specific frame regions, not prioritizing multi-event presentation for a single object. Methodologically, it uses local hard-masked attention and global cross-attention weighted addition, differing from TS-Attn's soft modulation approach. While effective for scene-level generation, Videotetris's untrained local sub-prompt cross-attention may struggle with fine-grained event-level prompts, risking temporal discontinuities, reduced video quality, and subject inconsistencies.
>
>     - **TALC [3]**: TALC focuses on multi-scene generation along the temporal axis using hard-masked cross-attention to limit interactions within each segment and its sub-prompts. While effective for multi-scene tasks, this isolation can cause background changes across segments, making it less suitable for multi-event prompts requiring consistent backgrounds and subjects. In contrast, TS-Attn uses soft bindings between motion regions and events, ensuring visual consistency and event responsiveness in multi-event scenarios.
>
> To further demonstrate the advantages of the proposed TS-Attn in multi-event generation, we also extend the original TALC and VideoTetris frameworks as baselines. To better adapt them for multi-event generation, their required sub-prompts are replaced with prompts for each event and are applied sequentially according to their respective conditioning strategies.
>
> - **Experiment:**
>
> We apply TALC [2] and VideoTetris [3] on Wan2.2-A14B following the optimal hyperparameters described in their original papers, to ensure a fair comparison with TS-Attn. As shown in **Table A**, TS-Attn significantly outperforms TALC and VideoTetris. TALC's restrictive conditioning of each segment on sub-prompts compromises global coherence, reducing performance. While VideoTetris incorporates weighted global and local cross-attention, the lack of training distorts the original video latent distribution, degrading quality and offering limited improvement. We also present qualitative visual comparisons in Figure 18 (Appendix).
>
> **Table A. More comparison with prior methods**
> | Model         | Human | Animal | Object | Retrieval | Creative | Easy  | Hard  | Average |
> |---------------|-------|--------|--------|-----------|----------|-------|-------|---------|
> | Wan2.2-A14B   | 51.2% | 46.7%  | 44.9%  | 54.8%     | 34.8%    | 60.3% | 34.0% | 48.3%   |
> | + TALC        | 50.9% | 45.4%  | 44.1%  | 56.2%     | 33.8%    | 60.6% | 31.9% | 47.1%   |
> | + VideoTertis | 53.0% | 46.5%  | 46.8%  | **63.6%**     | 35.9%    | 63.5% | 37.5% | 49.7%   |
> | **+ Ours**    | **60.4%** | **53.6%**  | **52.0%**  | 63.0%     | **45.3%**    | **70.5%** | **44.3%** | **56.2%**   |
>
> [1] DreamRunner: Fine-Grained Compositional Story-to-Video Generation with Retrieval-Augmented Motion Adaptation, 2024.
>
> [2] TALC: Time-Aligned Captions for Multi-Scene Text-to-Video Generation, 2024.
>
> [3] VideoTetris: Towards Compositional Text-to-Video Generation, 2024.

---

> ### Author Response · Authors · 2025-11-24
> **Response to Reviewer Qohm (2/2)**
>
> > ### **Weakness 2: Generalization to multi-subject scenarios**
>
> Thanks for your question!
>
> - The analysis and formulation of applying TS-Attn to multiple subjects has been briefly presented in Appendix A. We would like to highlight that in the presence of multiple subjects, each is assigned a dedicated motion region mask to facilitate the grounding and attention modulation between multiple visual regions and the event list. Additionally, we would like to reference the prompt mentioned by the reviewer here to elaborate on the processing flow in such cases.
>
> - Specifically, "leaves" and "enters" are temporally assigned to the two parts of the frames, respectively, with their corresponding subjects being "person" and "cat." We construct an attention rearrangement mask to achieve two goals: 1) For the first part of the frames, video tokens associated with the "person" layout enhance attention to "leaves" while suppressing attention to "enters," with the remaining video tokens preserving the original attention distribution; 2) For the second part of the frames, video tokens associated with the "cat" layout enhance attention to "enters" while suppressing attention to "leaves," with the remaining video tokens maintaining the original attention distribution.
>
> - This approach binds the temporally relevant subjects to their corresponding events, while the unmodulated video tokens interact with all text tokens, leveraging the baseline model's capability to handle potential subject transitions. For more examples involving multiple subjects, please refer to **Figure 19**, where we also provide illustrative examples of mask construction to better visualize the process.

---

### Official Review · Reviewer_HkQf · 2025-11-04

**Soundness:** 3
**Presentation:** 3
**Contribution:** 3
**Rating:** 6
**Confidence:** 5

**Summary:**

the paper proposes a training-free temporal-wise separable attention for multi-event conditioned video generation

**Strengths:**

- the motivation of the work is clear and well-explained. the proposed motion region extraction and event-based attention modulation are intuitive and visualizations are reasonable
- extensive experiments demonstrate the effectiveness of the proposed method. the visual results are convincing and clearly show the multi-event coherence
- the method achieves reasonable performance improvements with negligible overhead
- the paper provided source codes in the supplementary material

**Weaknesses:**

- the benchmark protocol and the evaluation metrics are not well-justified. while I understand there is no reasonable benchmark framework in the current field, the reliability of the vlm-based evaluation is still questionable, especially when evaluated against a commercial endpoint. in that case, it is hard to determine the actual performance gain based on the reported scores. a user study is highly recommended to validate the effectiveness of the proposed method considering the human evaluation is still the most reliable metric for video generation tasks
- it is unclear what is the max possible number of events the proposed method can handle
- what is the success rate of a given prompt? what are the typical failure cases?
- while the proposed framework provides a solution for multi-subjects, it is unclear whether the proposed framework can faithfully handle multiple subjects with same/similar actions

**Questions:**

please refer to the weaknesses section

---

> ### Author Response · Authors · 2025-11-24
> **Response to Reviewer HkQf (1/4)**
>
> We sincerely thank you for taking the time to review our paper and providing valuable feedback. We are delighted to know that the motivation behind our method and the experimental setup have been positively received by you. We will now proceed to address your comments in detail：
>
> > ### **Weakness 1: Conduct user study to evaluate performance**
>
> We first thank the reviewer for this valuable concern. We conducted a user study based on a paired voting strategy. Each questionnaire was designed to randomly select 50 samples from all the evaluation sets, with participants asked to judge which video is better or if it is a tie. The paired videos were randomly drawn from different models. For each question, participants primarily made their selections based on the completion of each event and the overall temporal consistency. To ensure comprehensive coverage of the evaluation, we recruited 10 experienced participants to capture a broader range of subjective opinions. The final summary results are shown in the **Table A** below, which indicate that our method significantly outperforms existing state-of-the-art models, validating the effectiveness of our design in multi-event video generation tasks.
>
> **Table A: Human preference study.**
> | Model                      | Others win | Tie  | Ours win |
> |----------------------------|-----------------|------|---------------|
> | SkyReels-V2-14B             | 16%             | 30%  | 54%           |
> | MAGI-1-24B                  | 12%             | 28%  | 60%           |
> | Wan2.2-A14B                 | 24%             | 36%  | 40%           |
> | MEVG + Wan2.2-A14B          | 8%             | 20%  | 72%           |
> | DiTCtrl + Wan2.2-A14B       | 10%             | 24%  | 66%           |

---

> ### Author Response · Authors · 2025-11-24
> **Response to Reviewer HkQf (2/4)**
>
> > ### **Weakness 2: Max possible number of events**
>
> - Thank you for your question! The maximum number of events that TS-Attn can handle depends on the number of seconds supported by the video foundation models for generation. Specifically, wan2.1 and wan2.2 support generating 5-second videos at 16 fps. Therefore, the recommended maximum number of allowable events is approximately **5**, ensuring that each event has sufficient time to be presented while avoiding overly rapid transitions between events that could cause motion anomalies.
>
> - Generating more events can be achieved by applying TS-Attn to frameworks that support video continuation, such as the recently released LongCat-Video-13.6B model [1]. For instance, each clip can generate 3 events, and by continuing N clips, a total of **3N** events can be produced. This approach ensures that each event has sufficient time for presentation while maintaining good temporal consistency. Detailed experiments on applying TS-Attn to LongCat-Video are presented in **Weakness 3**.
>
> [1] Meituan team. "LongCat-Video Technical Report". https://arxiv.org/abs/2510.22200

---

> ### Author Response · Authors · 2025-11-24
> **Response to Reviewer HkQf (3/4)**
>
> > ### **Weakness 3: More explanation on success rate & Failure case analysis**
>
> Thank you for your question! We elaborate on these two questions in detail below.
>
> - **Success rate:**
>     - We generated or sampled 30 prompts for each containing 3 events, 4 events, 5   events, 6 events, totaling 120 prompts. Ten experienced participants were recruited to evaluate the success rate of generation for each prompt. If the majority of participants agree that the video accurately presents all events, the generation is considered successful; otherwise, it is deemed a failure.
>
>     - As shown in **Table A** below, TS-Attn demonstrates a relatively high success rate for 3-4 events, achieving 73.3% and 46.7%, respectively, while the baseline model shows significantly lower success rates of 43.3% and 23.3%. When the number of events increases to 5-6, the success rate of TS-Attn drops from 30.0% to 13.3%. The primary reason for this decline lies in the limited video duration supported by the Wan2.2 baseline model. When the number of events exceeds 5, it becomes challenging to fully present each event within the allocated frames.
>
>
> - **Failure case analysis:**
>     - As discussed above, failure cases primarily occur when the number of events far exceeds the video duration supported by foundation models, leading to issues such as event omission and unnatural motion transitions. This is mainly due to the short-clip generation paradigm adopted by models like Wan2.2, Wan2.1, and CogVideoX. A potential solution to this issue is to leverage recently released model frameworks that natively support continuous video generation, such as LongCat-Video [1].
>
>     - In this case, we can distribute a large number of events across multiple clips. For example, 9 events can be divided into 3 clips for completion while ensuring temporal consistency. To validate this, we further create prompts containing 7, 8, and 9 events, with 30 prompts for each case, and divided them into 3 clips for generation (i.e., 2+2+3=7, 2+3+3=7, 3+3+3=9). During the generation of each clip, we apply TS-Attn and compared its performance with the baseline.
>
>     - As shown in **Table B**, TS-Attn enhances temporal awareness within each clip, significantly improving the overall ability to handle videos with a large number of events. The advantages of applying TS-Attn to architectures like LongCat-Video are twofold: 1) Given a fixed number of events, TS-Attn can complete generation with fewer clips; 2) Given a fixed number of clips, TS-Attn can robustly handle more complex temporal descriptions. This further demonstrates the potential of TS-Attn for interactive video generation and long video generation.
>
>     - Other minor failure cases mainly occur when the user-provided prompt is temporally illogical, potentially causing temporal errors. This can be addressed by incorporating an LLM API as a prompt rewriter to refine the prompts.
>
>
> **Table A: The variation in success rate with increasing number of events.**
> | Model      | 3 events | 4 events | 5 events | 6 events |
> |------------|----------|----------|----------|----------|
> | Wan2.2-A14B | 43.3%    | 23.3%    | 16.7%    | 0%       |
> | +TS-Attn    | 73.3%    | 46.7%    | 30.0%    | 13.3%    |
>
>
> **Table B: The variation in success rate with a large number of events. Three clips are generated for each prompt.**
> | Model               | 7 events | 8 events | 9 events |
> |---------------------|----------|----------|----------|
> | LongCat-Video-13.6B  | 26.7%    | 13.3%    | 6.7%     |
> | +TS-Attn             | 53.3%    | 40.0%    | 33.3%    |
>
>
> [1] Meituan team. "LongCat-Video Technical Report". https://arxiv.org/abs/2510.22200

---

> ### Author Response · Authors · 2025-11-24
> **Response to Reviewer HkQf (4/4)**
>
> > ### **Weakness 4: Handle multiple subjects with same or similar actions**
>
> We appreciate your question!
> We have added examples of multiple subjects performing the same events **(the third case of Figure 19)**. It can be observed that TS-Attn accurately binds the two events, "running" and "stop to rest," to both the "man" and the "woman," while ensuring temporal accuracy. In contrast, the baseline model exhibits event omission (the woman does not run). We have also presented more examples of multi-subject multi-event generation in **Figure 19**, along with corresponding attention rearrangement mask diagrams to provide a more intuitive illustration of the TS-Attn mechanism.

---

### Meta-Review · Area_Chair_uPwm · 2025-12-27

**Summary:**

The paper aims to improve multi-event generation for pre-trained video generation models by introducing a training-free approach. The reviewers raised questions and concerns, including the need for more technical details (e.g., discussion of motion region extraction and temporal segmentation), further explanation and results for challenging scenarios (e.g., multiple subjects and sudden scene changes), and additional experimental results (e.g., user studies, the maximum possible number of generated events, analysis of success and failure cases, and comparisons with existing works). The authors provided detailed responses to each question and concern. Three out of four reviewers gave positive ratings, with one reviewer further increasing their score and confirming that their concerns had been addressed. Overall, the feedback from the reviewers on the paper is positive.

**Reviewer Concerns:**

In the rebuttal, the authors provided very detailed responses to each concern and question raised by the reviewers. I believe the rebuttal has addressed all of the concerns.

**Reviewer Scores:**

During the limited discussion period, only Reviewer oX7g participated in the discussion and indicated that the raised concerns had been addressed. Reviewers HkQf and 3mMY gave positive ratings, and the authors provided detailed responses to their concerns and questions. These two reviewers are likely to maintain their positive ratings. Reviewer Qohm gave a negative rating before the rebuttal, with major concerns regarding comparisons with prior works and generalization to multi-subject generation. The authors presented quantitative and qualitative results to address these concerns. The reviewer may lean toward a positive rating after engaging in the discussion.

---

### Decision · Program_Chairs · 2026-01-26

Accept (Poster)